# Enabling new insights from old scans by repurposing clinical MRI archives for multiple sclerosis research

Philipp Goebl [1,2] ✉, Jed Wingrove[1], Omar Abdelmannan[1], Barbara Brito Vega [1,2], Jonathan Stutters[1], Silvia Da Graca Ramos[3], Owain Kenway[3], Thomas Rossor[4], Evangeline Wassmer [5,6], Douglas L. Arnold [7], D. Louis Collins [7], Cheryl Hemingway[8], Sridar Narayanan [7], Jeremy Chataway[1,9], Declan Chard [1,9], Juan Eugenio Iglesias[2,10,11], Frederik Barkhof [1,2,12], Geoff J. M. Parker[2,13,14], Neil P. Oxtoby [2], Yael Hacohen[1], Alan Thompson [1,9], Daniel C. Alexander [2,15], Olga Ciccarelli [1,9] & Arman Eshaghi [1,2]

Magnetic resonance imaging (MRI) biomarkers are vital for multiple sclerosis (MS) clinical research and trials but quantifying them requires multi-contrast protocols and limits the use of abundant single-contrast hospital archives. We developed MindGlide, a deep learning model to extract brain region and white matter lesion volumes from any single MRI contrast. We trained MindGlide on 4247 brain MRI scans from 2934 MS patients across 592 scanners, and externally validated it using 14,952 scans from 1,001 patients in two clinical trials (primary-progressive MS and secondary-progressive MS trials) and a routine-care MS dataset. The model outperformed two state-of-the-art models when tested against expert-labelled lesion volumes. In clinical trials, MindGlide detected treatment effects on T2-lesion accrual and cortical and deep grey matter volume loss. In routine-care data, T2-lesion volume increased with moderate-efficacy treatment but remained stable with high-efficacy treatment. MindGlide uniquely enables quantitative analysis of archival single-contrast MRIs, unlocking insights from untapped hospital datasets.

Multiple sclerosis (MS) is a chronic disabling disease affecting over 2.8 million people worldwide, with a disproportionate impact on young populations. Magnetic resonance imaging (MRI) biomarkers are central to MS phase 2 and 3 clinical trials as primary and secondary efficacy endpoints. Typical imaging protocols include multi-contrast MRI scans to capture distinct aspects of disease evolution: new or enlarging lesions indicate active inflammation on fluid-attenuated inversion recovery (FLAIR) and T2-weighted images. In contrast, brain atrophy on T1-weighted images is a proxy for neurodegeneration. However, multi-contrast acquisitions are time-consuming, costly, and less available in the clinical setting. Simplifying MRI analysis, particularly

through single-contrast brain volume calculations, can expand research opportunities from hospital archives, allowing analysis of previously acquired routine-care scans and potentially making clinical trials less costly by reducing the need for multi-contrast acquisitions.

Although routine MRI monitoring in MS care primarily tracks new or enlarging lesions, untapped potential lies in leveraging this readily available data also to assess brain atrophy, a key contributor to disability worsening often overlooked in standard clinical practice. Unlike the standardised MR acquisition protocols used in pharmaceutical clinical trials, archived routine clinical scans are more heterogeneous and present a major obstacle to the reliable, automated volumetry of

---

brain structures and lesions. We aimed to develop a computationally efficient tool to process these highly heterogeneous scans independent of contrasts, resolutions, and qualities. Such a tool would extract key MRI biomarkers in diverse clinical settings, enable real-world research and pave the way for automatic analysis to aid clinical decision-making.

Recent advances in deep learning and generative models have enhanced our ability to analyse routine-care MRI scans. These advances include tools like SynthSeg and others, primarily focusing on brain segmentation after removing (or inpainting) lesions[1-6]. SAMSEG-lesion is a newly introduced model that robustly segments lesions and brain structures across MRI contrasts[4]. A model named WMH-SynthSeg can segment white matter hyperintensities and brain anatomy simultaneously from scans of varying resolutions and contrasts, including low-field portable MRI[7]. However, these tools were not developed or validated to assess treatment effects within the diverse and heterogeneous MS routine care data and clinical trials.

Two recent studies demonstrated the feasibility of deep-learning-based quantification of thalamic and lesion volumes in routine clinical MRI scans for MS patients[8,9]. However, these studies did not evaluate treatment effects or include the wide variety of clinical-grade two-dimensional scans of different contrasts (T1-weighted, T2-weighted, and T2-FLAIR contrasts) and scans from clinical archives. This highlights the urgent need for solutions that extract MRI biomarkers, including lesion load and changes in brain volume, from the varying scans acquired in routine care for research repurposing and potential future clinical applications.

Here, we present MindGlide, a publicly available deep-learning model that addresses these limitations. We aimed to (1) efficiently (in less than a minute with no pre-processing required by the user) quantify brain structures and lesions from varying single MRI contrast inputs; (2) detect brain volume changes due to treatment effects using MRI contrasts not typically analysed for these purposes (such as T2 scans), and (3) demonstrate the potential of routine MRI scans to detect new lesions and subtle brain tissue loss, even when the ideal imaging contrasts are unavailable across age ranges.

## Results

### Study overview and patient characteristics

We developed MindGlide, a 3D convolutional neural network (CNN), to segment specific brain structures and white matter lesions. As Fig. 1 shows, MindGlide processes brain MRI across commonly available MRI contrasts in hospital archives and associated tissue intensities (T1-weighted, T2-weighted, proton density (PD), and T2-Fluid Attenuated Inversion Recovery [FLAIR]), including both 2D (two-dimensional) and 3D (three-dimensional) scans.

We used distinct training and external validation datasets consisting exclusively of patients with MS. Table 1 summarises the patient characteristics of our training and validation datasets. The training set comprised seven previously published clinical trials of relapsing-remitting (RR), secondary progressive (SP) and primary progressive (PP) MS[10-18] and one observational cohort (see Supplementary Table 1 for the list of clinical trials). During model training, we used 4247 real MRI scans (2092 T1-weighted, 2155 FLAIR) from 2871 patients with the following subtypes: SPMS (n = 1453), RRMS (n = 1082), PPMS (n = 336). These scans were acquired from 592 MRI scanners from 1.5 and 3 tesla magnetic fields. We generated 4303 synthetic scans to augment training and trained MindGlide on a dataset of 8550 real and synthetic images. The training set included only T1-weighted and FLAIR MRI contrasts and synthetically generated scans. We froze model parameters after training completion.

To test MindGlide's generalisability across ages 14–64, we employed an external validation set of two progressive MS clinical trials[16,19-24] and a real-world cohort of paediatric relapsing-remitting MS patients. This set encompassed T2-weighted and PD MRI contrasts and

T1-weighted and FLAIR contrasts. We specifically selected the paediatric cohort to test robustness across age groups, given the typically more inflammatory disease course and larger lesion volume relative to skull size in early-onset cases. Our external validation dataset consisted of 1001 patients from 186 MRI scanners, including the PPMS trial (n = 699), SPMS trial (n = 141), and routine-care paediatric RRMS cohort (n = 161)[20,24]. The PPMS dataset comprised 11,015 MRI scans (2756 T1-weighted, 2754 T2-weighted, 2749 FLAIR, 2756 PD), all with a slice thickness of 3 mm (1 mm × 1 mm × 3 mm). The SPMS dataset included 763 scans (378 T1-weighted, 385 T2-weighted) with varying slice thicknesses (T1: 1 mm isotropic, T2: 3 mm × 1 mm × 1 mm). The real-world paediatric cohort consisted of 161 individuals with 1478 scans (523 T1-weighted, 475 T2-weighted, 480 FLAIR) and diverse slice thicknesses (median: 3.3 mm, range: 0.4–8.5 mm). The median follow-up time was 28 months (standard deviation or SD: 8 months) in the PPMS dataset, 26 months (SD: 7 months) in the SPMS dataset, and 12 months (SD: 19 months) in the real-world paediatric cohort. In the real-world cohort, 89 received moderate efficacy and 72 received high efficacy treatments.

### Cross-sectional validation in external cohorts

**Expert-labelled lesions and correlations of segmented volumes with disability.** Throughout the manuscript, we refer to WMH-SynthSeg and SAMSEG as state-of-the-art. Figure 2 shows the results of comparing MindGlide with state-of-the-art in various metrics. We calculated the degree of agreement (dice score) of MindGlide-derived, SAMSEG-derived, and WMH-Synthseg-derived volumes with ground truth hand-labelled lesions on the same scans (Fig. 2b). The median Dice score was 0.606 for MindGlide, 0.504 for SAMSEG and 0.385 for WMH-Synthseg. Supplementary Table 2 summarises the cross-software comparison of lesion segmentations (MindGlide, SAMSEG and WMH-Synthseg)[4,7].

As Fig. 2c shows, in our PPMS dataset, MindGlide-derived lesion load had a numerically higher correlation with the Expanded Disability Status Scale (EDSS) than the state-of-the-art. However, the differences were not statistically significant for all comparisons. When using T2-FLAIR scans, the correlation between MindGlide-derived lesion load and the EDSS was on average higher (correlation coefficient = 0.127, P < 0.001) than the correlations observed with SAMSEG (correlation coefficient = 0.009, P = 0.813) or WMH-Synthseg (correlation coefficient = 0.105, P = 0.005). When using T2 scans, MindGlide showed significant correlation between lesion load and EDSS (correlation coefficient = 0.150, P < 0.001), which was also the case for SAMSEG (correlation coefficient = 0.086, P = 0.022) and WMH-Synthseg (correlation coefficient = 0.140, P < 0.001). In head-to-head comparisons, we observed a significant difference between the correlation coefficients of MindGlide and SAMSEG lesion loads from FLAIR (P = 0.026), but not for T2 (P = 0.227). There was no difference between MindGlide and WMH-Synthseg lesion loads for either contrast (FLAIR: P = 0.680; T2: P = 0.850). As a reference, the correlation coefficient between ground truth expert-labelled hyperintense T2 lesion volumes and EDSS was 0.131 (95% CI: 0.057–0.203; P < 0.001).

Regarding the correlation between deep grey matter (DGM) volumes and EDSS scores, MindGlide-derived DGM volume demonstrated a correlation coefficient of −0.130 on FLAIR contrasts (95% CI: −0.202 to −0.057; P < 0.001), −0.128 on T2-weighted (95% CI: −0.200 to −0.054; P < 0.001) and −0.131 on 2D T1-weighted images (95% CI: −0.203 to −0.057; P < 0.001). Conversely, SAMSEG-derived DGM volume yielded correlation coefficients of −0.057 (95% CI: −0.130 to 0.018; P = 0.134), −0.084 (95% CI: −0.157 to −0.010; P = 0.026) and −0.106 (95% CI: −0.178 to −0.031; P = 0.005) for FLAIR, T2-weighted and 2D T1-weighted contrasts, respectively. WMH-Synthseg-derived DGM volume yielded correlation coefficients of −0.106 (95% CI: −0.179 to −0.032; P = 0.005), −0.112 (95% CI: −0.185 to −0.038; P = 0.003) and −0.112 (95% CI: −0.188 to −0.035; P = 0.004) for FLAIR, T2-weighted and T1-weighted contrasts. There were no statistically significant

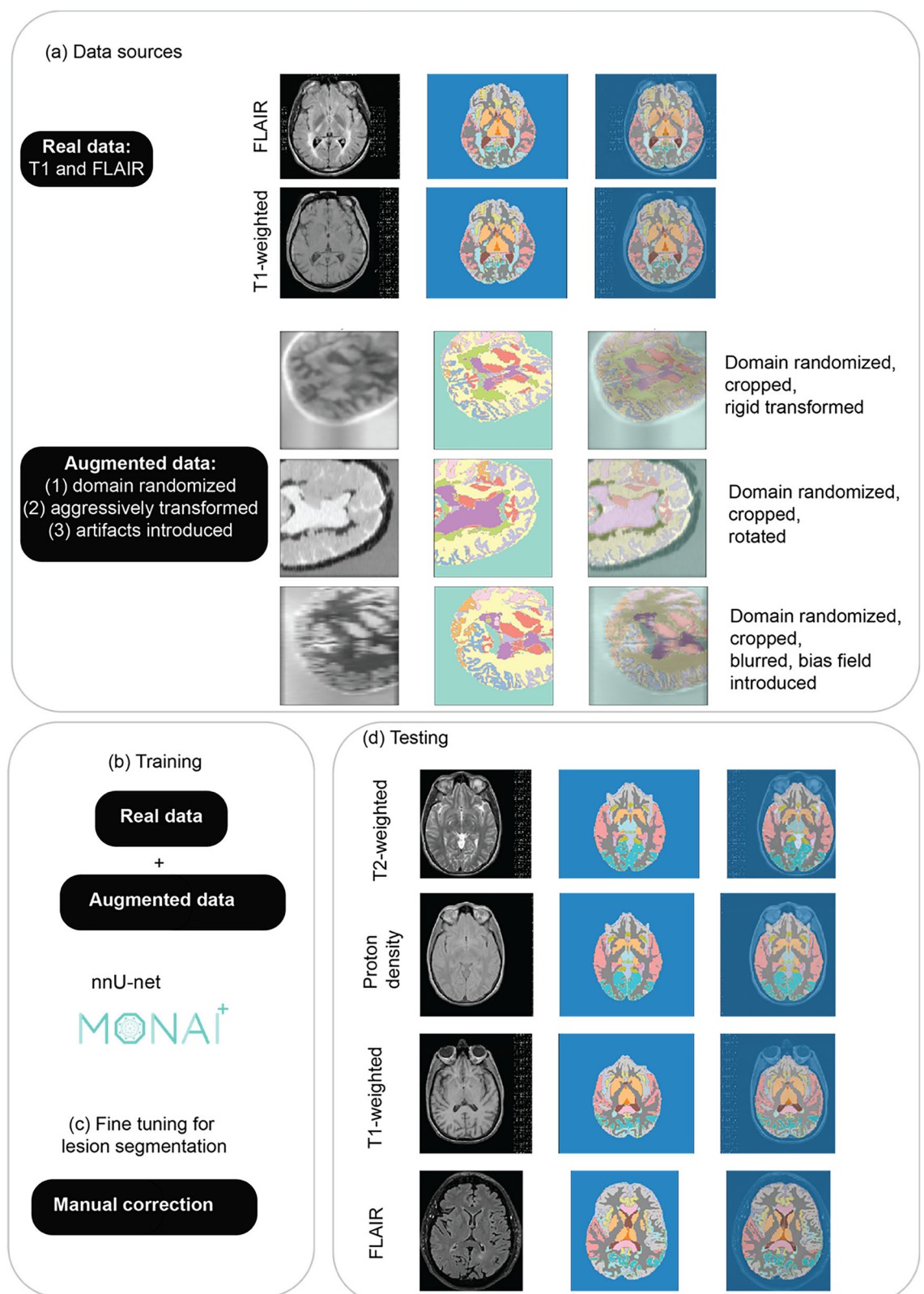

**Fig. 1 | Developing and testing the MindGlide model.** MindGlide model enables highly efficient and robust MRI segmentation. Segmenting and quantifying lesions on heterogeneous contrasts with minimal pre-processing (and no pre-processing required by the user). MindGlide model generalizes to tasks not used to train the model, such as segmenting T2-weighted and positron density MRI scans in unseen data sets. **a** Provides an overview of real (top) and augmented (bottom) training data. **b**, **c** Illustrate all parts of our training and fine-tuning pipeline. **d** Shows images of heterogenous contrasts used for testing MindGlide. FLAIR Fluid Attenuated Inversion Recovery, MRI magnetic resonance imaging.

**Table 1 | Patient characteristics in the training and external validation sets**

| Training and model development set | | | |
|---|---|---|---|
| | RRMS | SPMS | PPMS |
| Number (percentage of the sample) | 1082 (38%) | 1453 (51%) | 336 (12%) |
| Age (±standard deviation) | 37.1 ± 9.8 | 47.8 ± 8.2 | 49.8 ± 8.2 |
| Female (%) | 70.6% | 60.6% | 45.8% |
| Disease duration (IQR) | 1.2 (0.3–4.5) | 11.7 (5.5–16.5) | 5.0 (2.5–10.8) |
| EDSS (IQR) | 2.0 (1.5–3.5) | 6.0 (5.0–6.5) | 4.5 (4.0–6.0) |
| External validation set | | | |
| *SPMS (n = 141)* | | | |
| Age (± standard deviation) | – | 51.3 ± 6.3 | – |
| Female (%) | – | 69.3% | – |
| Disease duration (IQR) | – | 20.6 (15.3–27.5) | – |
| EDSS (IQR) | – | 6.0 (5.5–6.5) | – |
| Follow up in months (SD) | – | 26 (7) | – |
| # of follow up MRIs | – | 2 | – |
| MRI resolution | – | 2D-T1: 1×1×3 mm<br>2D-T2: 1×1×3 mm<br>3D-T1: 1×1×1 mm | – |
| *PPMS (n = 699)* | | | |
| Age (±standard deviation) | – | – | 44.6 ± 8.0 |
| Female (%) | – | – | 49.8% |
| Disease duration (IQR) | – | – | 1.5 (0.5–4.0) |
| EDSS (IQR) | – | – | 4.5 (3.5–6.0) |
| Follow up in months (SD) | – | – | 28 (8) |
| # of follow up MRIs | – | 3 | – |
| MRI resolution | – | T1: 1×1×3 mm<br>T2: 1×1×3 mm<br>T2-FLAIR:<br>1×1×3 mm<br>PD: 1×1×3 mm | – |
| *Real world paediatric cohort (n = 161)* | | | |
| Age (±standard deviation) | 14.5 ± 0.2 | – | – |
| Female (%) | 77.8% | – | – |
| Disease duration (IQR) | 0.2 (0.1–1.0) | – | – |
| EDSS (IQR) | 1.0 (1.0–1.5) | – | – |
| Follow up in months (SD) | 12 (19) | – | – |
| Median MRI slice thickness (Min–Max) | 3.3 mm (0.4–8.5)<br>T1: 1.0 mm (0.4–7.7)<br>T2-FLAIR: 1.0 mm (0.4–7.2)<br>T2: 4.8 mm (1.0–8.5) | – | – |

In the external validation set, SPMS Trial is the MS-STAT trial and PPMS trial is the ORATORIO trial. See Supplementary Table 1 for details.
*EDSS* Expanded Disability Status Scale, *PPMS* Primary Progressive Multiple Sclerosis, *SPMS* Secondary Progressive Multiple Sclerosis.

differences across the correlation coefficients obtained using MindGlide, SAMSEG and WMH-Synthseg on either FLAIR ($P = 0.439$), T2-weighted ($P = 0.680$) or T1-weighted contrasts ($P = 0.885$).

In the cortical grey matter (CGM), MindGlide-derived volumes and EDSS correlation coefficients were −0.123 on FLAIR (95% CI: −0.195 to −0.049; $P = 0.001$), −0.102 on T2 (95% CI: −0.175 to −0.028; $P = 0.007$) and −0.135 on T1 contrasts (95% CI: -0.207 to -0.061; $P < 0.001$). SAMSEG-derived CGM volume and EDSS correlation coefficients were −0.121 on FLAIR (95% CI: −0.193 to −0.047; $P = 0.001$),

0.053 on T2 (95% CI: −0.021 to 0.126; $P = 0.160$) and −0.114 on T1 contrasts (95% CI: −0.186 to −0.039; $P = 0.002$). WMH-Synthseg-derived CGM volume yielded correlation coefficients of −0.091 (95% CI: −0.164 to −0.017; $P = 0.016$), −0.114 (95% CI: −0.186 to −0.040; $P = 0.026$) and −0.127 (95% CI −0.202 - −0.050; $P = 0.001$) for FLAIR, T2-weighted and T1-weighted contrasts. There was a statistically significant difference between the correlation coefficients when using MindGlide or SAMSEG on T2 contrasts ($P < 0.001$) but not on FLAIR ($P = 0.966$) or T1 ($P = 0.691$). There was no difference between correlation coefficients obtained using MindGlide and WMH-Synthseg (FLAIR: $P = 0.549$; T2: $P = 0.828$; T1: $P = 0.879$).

## Longitudinal validation
**Treatment effects on lesion accrual.** When we tested longitudinal lesion accrual in the SPMS trial (simvastatin vs placebo) using T2-weighted MRI, the rate of MindGlide-derived lesion volume accrual was not significantly faster in the placebo group than in the treatment group (1.12 mL/year vs 0.768 mL/year; $P = 0.054$)[24]. Using 3D T1-weighted MRI, the rate of hypointense lesion accrual was significantly faster in the placebo group than in the treatment group (1.874 mL/year vs 1.071 mL/year; $P = 0.005$).

As Fig. 3 shows the PPMS trial (ocrelizumab vs. placebo) had a slower MindGlide-derived lesion volume accrual rate in the treatment group across all MRI contrasts: T2-weighted (1.103 mL/year vs. 0.399 mL/year), FLAIR (1.042 mL/year vs. 0.141 mL/year), PD (0.633 mL/year vs. 0.91 mL/year) hyperintense lesions, and T1-weighted hypointense lesions (1.225 mL/year vs. 0.648 mL/year). All differences were statistically significant ($P < 0.001$).

As Fig. 4 shows, MindGlide-derived lesion volumes in the T2 scans of the routine-care paediatric cohort increased by 0.612 ml/year in the moderate-efficacy treatment group ($P < 0.001$), while remaining stable in the high-efficacy treatment group (model-estimated average −0.376 ml/year, $P = 0.230$). FLAIR lesion volumes were stable in both groups: −0.063 ml/year in the high-efficacy group ($P = 0.824$) vs 0.009 ml/year in the moderate efficacy group ($P = 0.966$). T1-weighted hypointense lesion volumes remained stable in the high-efficacy group (model-estimated average 0.165 ml/year, $P = 0.611$) but increased by 0.647 ml/year in the moderate-efficacy group ($P = 0.001$).

## Treatment effects on brain tissue loss
Figure 5 illustrates example segmentations across different contrasts and Table 2 shows the treatment effects in the two clinical trials. In the SPMS trial (simvastatin vs. placebo), MindGlide showed a significantly slower rate of cortical GM volume loss in the treatment group than placebo using MRI contrasts previously unused for this purpose such as T2-weighted MRI. This effect was consistent across both T2-weighted (−0.704 mL/year (95% CI [−1.254−0.155]) vs. −1.792 mL/year (95% CI [−2.089−−1.495]), $P = 0.008$) and 3D T1-weighted MRI (−1.630 mL/year (95% CI [−2.283−−0.976]) vs. −2.912 mL/year (95% CI [−3.266−−2.558]), $P = 0.009$. The DGM had a slower rate of loss in the treatment group compared to placebo for both T2-weighted (−0.102 ml/year (95% CI [−0.155−−0.048]) vs. −0.205 ml/year (95% CI [−0.234−−0.176]), $P = 0.009$) and 3D T1-weighted contrasts (−0.105 ml/year (95% CI [−0.159−−0.050]) vs. −0.234 ml/year (95% CI [−0.263−−0.204]), $P = 0.001$).

In the PPMS trial (ocrelizumab vs. placebo), MindGlide consistently showed a slower rate of cortical GM volume loss in the treatment group across T2-weighted (−1.638 ml/year (95% CI [−1.820−−1.457]) vs. −2.335 ml/year (95% CI [−2.606−−2.065]), $P < 0.001$), T2-FLAIR (−1.778 ml/year (95% CI [−2.033−−1.524]) vs. −2.342 ml/year (95% CI [−2.722−−1.963]), $P = 0.016$), and PD contrast (−1.683 ml/year (95% CI [−1.980−−1.386]) vs. −2.310 ml/year (95% CI [−2.752−−1.868]), $P = 0.021$), with a similar trend in 2D T1-weighted contrast (−2.183 ml/year (95% CI [−2.363−−2.002]) vs. −2.485 ml/year (95% CI [−2.753−−2.217]), $P = 0.06$).

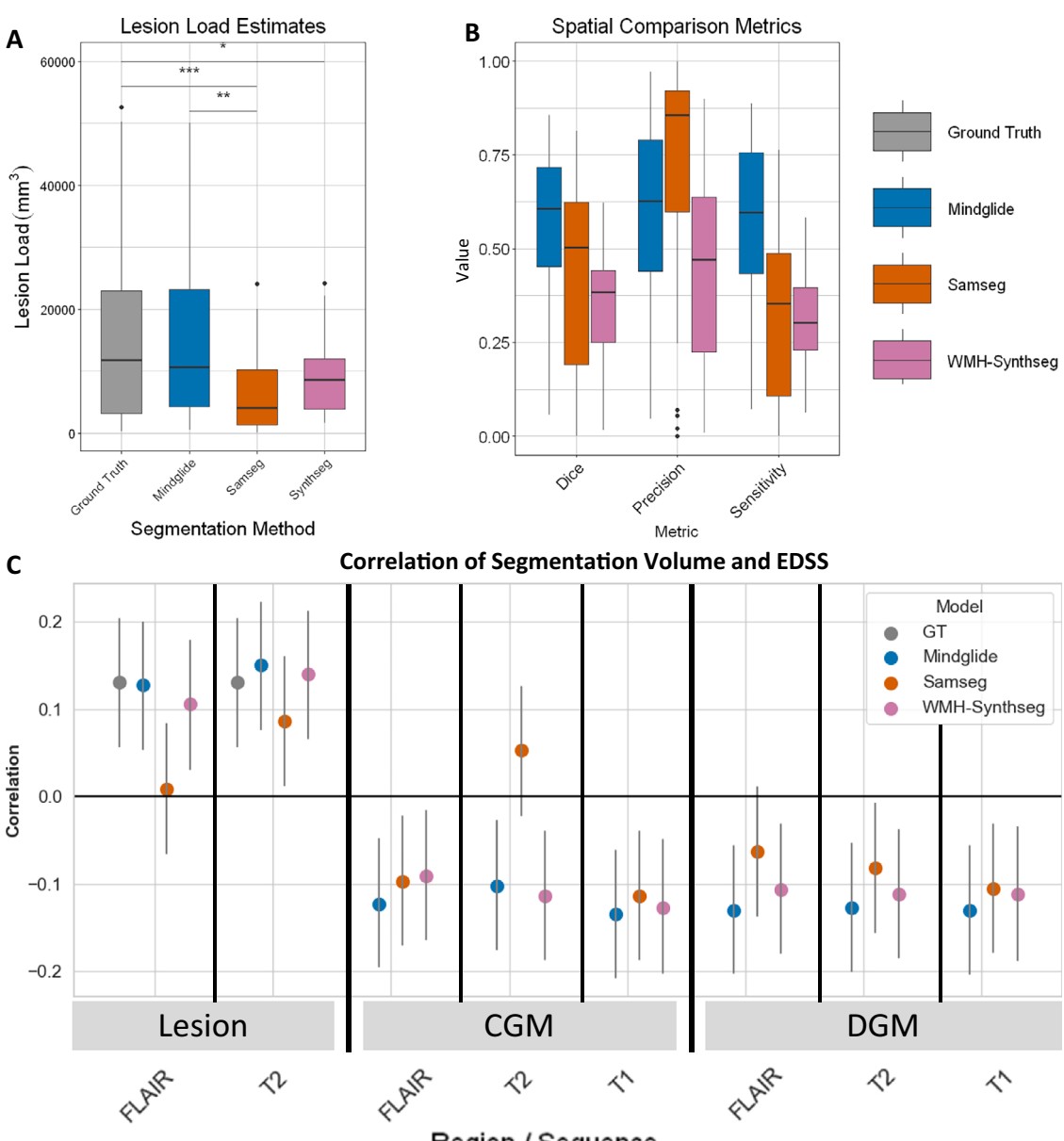

**Fig. 2 | Performance comparisons with state-of-the-art and ground truth.**
**A** Boxplot displaying Lesion Load estimates (mm³) and distributions measured using ground truth manual delineations (grey), MindGlide (blue) and Freesurfer's SAMSEG (orange). Lesion load estimates between Ground truth and SAMSEG and MindGlide and SAMSEG methods were significantly different (paired t-tests).
**B** Boxplot displaying Dice scores, Sensitivity and Precision measurements for both MindGlide (blue) and SAMSEG (orange) delineated lesions. ***$P < 0.001$, **$P < 0.01$, *$P < 0.05$. For (**A**) and (**B**) we used two openly available lesion segmentation data-sets comprising 50 brain MRI images and segmentation masks as ground truth comparators ($N = 50$, see Supplementary methods)[38,39]. In (**C**) we calculated Spearman's correlation coefficients for regional brain volumes obtained from MindGlide and Fressurfer's SAMSEG and WMH-Synthseg against the expanded disability status scale (EDSS). The analysis evaluates correlations of lesion, deep grey matter (DGM), and cortical grey matter (CGM) volumes with EDSS, across

FLAIR and T2 MRI contrasts. As a ground truth comparator for the correlation between lesion volume and EDSS we used manually labelled lesions by expert neuroradiologists. For all tested regions and contrasts MindGlide's output shows on average higher correlations with EDSS scores except for CGM in T2 (although as shown, they are not statistically significantly different across software). Error bars represent 95% CI. Data are presented as boxplots where the black line on the centre of the boxplot represents the median, the box encloses the lower and upper quartiles, and the whiskers extend to the minimum and maximum values within a range of 1.5 times the interquartile range. Values outside 1.5 times the interquartile range are displayed as black dots. For (**C**) we used the baseline images of our PPMS dataset ($N = 699$) and data are represented as Spearman's correlation coefficients and error bars indicate 95% confidence intervals. GT ground truth (manually labelled lesion segmentation by expert neuroradiologists). Source data are provided as a Source Data file.

Regarding deep GM volume loss, there was no significant difference between groups using T2-weighted MRI (−0.172 ml/year (95% CI [−0.203−−0.141]) vs. −0.143 ml/year (95% CI [[−0.164−−0.123]), $P = 0.130$). However, analysis with FLAIR (0.200 ml/year (95% CI [−0.234−−0.167]) vs. 0.156 ml/year (95% CI [−0.178−−0.133]), $P = 0.028$), PD (0.220 ml/year (95% CI [−0.255−−0.186]) vs. 0.144 ml/year (95% CI [−0.167−−0.121]), $P < 0.001$, and 2D T1-weighted images

(0.212 ml/year (95% CI [−0.245−−0.179]) vs. 0.172 ml/year (95% CI [−0.194−−0.150]), $P = 0.049$) consistently showed slower loss rates in the treatment group.

In the routine-care paediatric cohort, we observed cortical GM loss across all MRI contrasts with MindGlide segmentations in both treatment groups. T1 scans showed a loss of 5.807 ml/year (95% CI [−8.013−−3.601], $P < 0.001$) in the moderate-efficacy group and

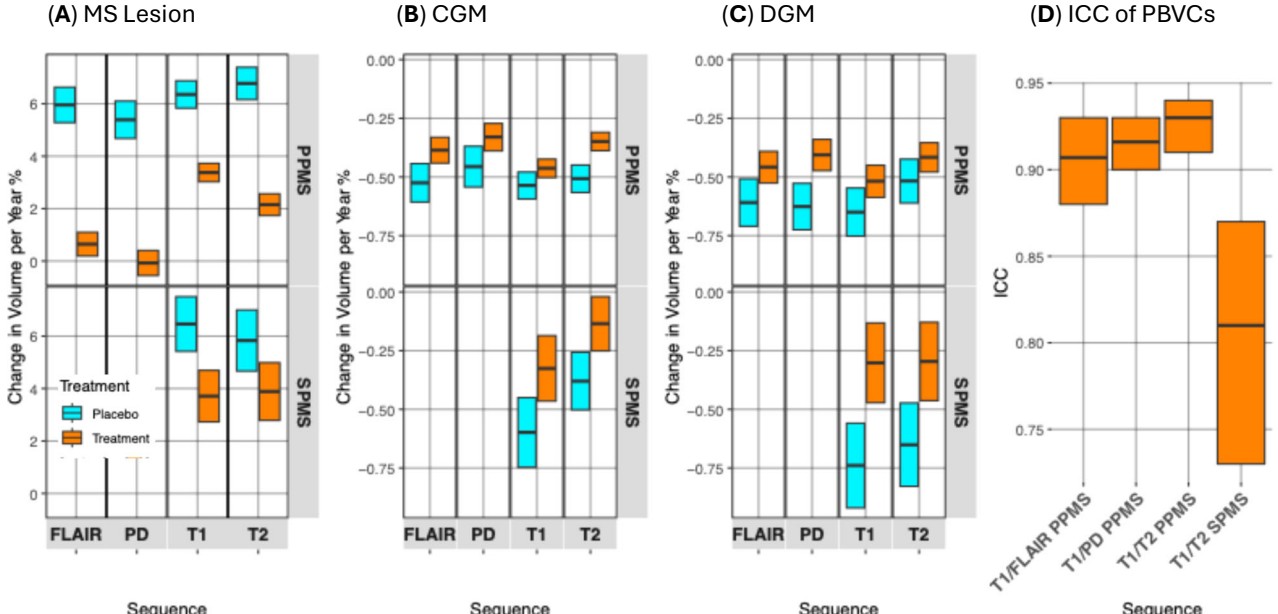

**Fig. 3 | Measuring treatment effects using single MRI contrasts.** MindGlide uniquely enables quantifying treatment effects using single MRI contrasts, including those that have never been used for this purpose (e.g., T2-weighted MRI). **A–C** shows longitudinal volume changes with results for our PPMS dataset on the top and results for our SPMS dataset on the bottom. **A** Illustrates the annual per cent change in lesion volume detected by MindGlide across FLAIR, PD, T1, and T2 contrasts (resolution: 1 × 1 × 3 mm) for primary progressive MS (PPMS) and secondary progressive MS (SPMS) cohorts, stratified by treatment allocation. Notably, treatment cohorts exhibited a reduction in lesion volume accrual compared to placebo across all contrasts. **B** Depicts the annualized rate of cortical grey matter (CGM) atrophy. MindGlide successfully differentiated between treatment and placebo groups, demonstrating reduced cortical atrophy across all MRI contrasts in

treated patients. This is also the case for atrophy rates in deep grey matter (DGM) as seen in (**C**). There are no FLAIR and PD contrasts available for the SPMS cohort. **D** Shows inter-contrast consistency for percentage brain volume changes (PBVC): High intra-class correlation coefficients (ICC) for percent brain volume change (PBVC) across different MRI contrasts within the PPMS dataset (2D), indicating high inter-contrast consistency. This underscores the segmentation tool's robustness and consistency in detecting neurodegenerative changes across various imaging contrasts. In the SPMS dataset we compared PBVC of 2D-T1 images and 2D-T2 images with an ICC-coefficient of 0.81 [95% CI 0.73–0.87]. PPMS: $N = 680$, SPMS: $N = 130$. All boxes in (**A–D**) display medians and 95% CI. Source data are provided as a Source Data file.

3.736 ml/year (95% CI [−7.423−−0.048], $P = 0.049$) in the high-efficacy group. T2-weighted images revealed a loss of 3.66 ml/year (95% CI [−4.401−−2.919], $P < 0.001$) and 2.102 ml/year (95% CI [−4.353-0.150], $P = 0.068$) respectively, while FLAIR scans showed a loss of 3.516 ml/year (95% CI [−5.626−−1.405], $P = 0.001$) and 4.19 ml/year (95% CI [−7.057−−1.322], $P = 0.005$).

Regarding deep GM, T1 contrast showed stable volume in the high efficacy group (-0.086 ml/year, (95% CI [−0.368-0.197], $P = 0.552$)) but a loss of 0.301 ml/year (95% CI [−0.474−−0.128], $P = 0.001$) in the moderate efficacy group. T2 images showed no change in the high efficacy group (−0.163 ml/year (95% CI [−0.416-0.090], $P = 0.207$)) but a loss of 0.32 ml/year (95% CI [−0.403−−0.236], $P < 0.001$) in the moderate efficacy group. FLAIR scans revealed a loss of 0.22 ml/year (95% CI [−0.421−−0.018], $P = 0.034$) and 0.399 ml/year (95% CI [−0.548−−0.250], $P < 0.001$) in the high and moderate efficacy groups, respectively.

In the power analysis single-contrast sample sizes were feasible for some, but not all, of the acquisitions. For example, for the PPMS group using only T2-weighted contrast and hyperintense lesion accrual as the primary outcome the sample size was 94, and for the cortical GM as the primary outcome a sample of 420 patients was required to achieve 80% statistical power. Supplementary Table 5 shows the complete sample size results.

### Comparing treatment effects with MindGlide against other segmentation tools and ground truth lesions in the PPMS clinical trial

In our analysis, MindGlide-derived lesion volumes demonstrated a treatment effect of 5.31% (95% CI [4.50–6.12%], $P < 0.001$) difference

between treatment groups in FLAIR images and 4.62% (95% CI [3.88–5.37%], $P < 0.001$) in T2 images (Fig. 6). This closely aligns with ground truth values, which indicated a 4.63% (95% CI [3.73–5.54%], $P < 0.001$) difference between treatment groups. In contrast, long-itudinal SAMSEG overestimated the treatment effect, showing a 10.70% (95% CI [3.42–17.98%], $P = 0.004$) difference for FLAIR images and 8.81% (95% CI [3.64–13.97%], $P = 0.001$) for T2 images, while WMH-Synthseg underestimated the effect with only a 2.56% (95% CI [1.64–3.47%], $P < 0.001$) difference in FLAIR images and 2.45% (95% CI [1.69–3.22%], $P < 0.001$) in T2 images. Additionally, longitudinal SAMSEG exhibited a broader spread in lesion changes across both T2 and FLAIR images compared to MindGlide-derived volumes, suggesting a higher precision in MindGlide's lesion volume estimation.

### Direct comparisons of tissue volumes and visual inspection in the PPMS trial and real-world cohort

As Fig. 6 shows, in assessing regional brain volumes, MindGlide-derived measurements demonstrated greater treatment effects between treatment groups compared to those obtained from long-itudinal SAMSEG or WMH-Synthseg. Specifically, MindGlide-derived CGM volume changes revealed a 0.14% (95% CI 0.04–0.24%], $P = 0.006$) difference between treatment groups for FLAIR images and a 0.16% (95% CI 0.09–0.23%], $P < 0.001$) difference for T2 images. In comparison, longitudinal SAMSEG-derived CGM volume changes indicated a 0.04% (95% CI −0.18–0.24%], $P = 0.744$) difference for FLAIR images and 0.08% (95% CI −0.07–0.24%], $P = 0.288$) for T2 images, while WMH-Synthseg-derived CGM volume changes exhibited a 0.11% (95% CI 0.02–0.19%], $P = 0.014$) difference for FLAIR images and a 0.12% (95% CI 0.05–0.2%], $P = 0.002$) difference for T2 images. For

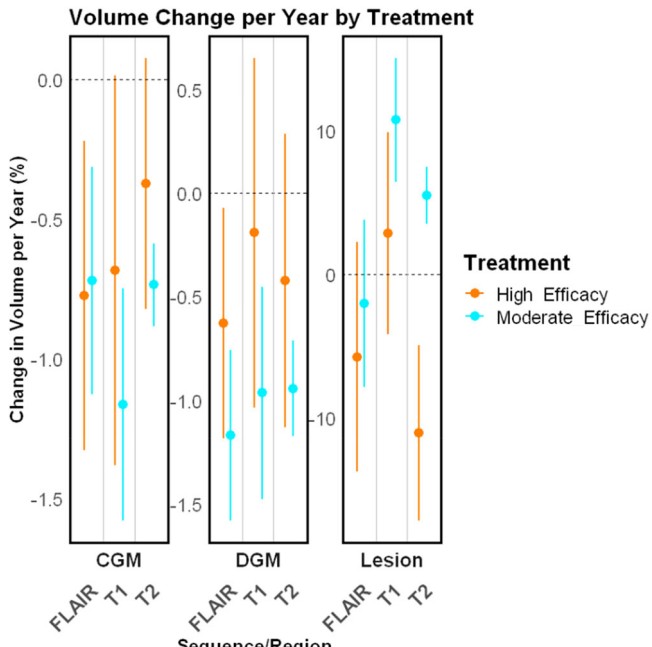

**Fig. 4 | Longitudinal changes of brain regions and lesion volumes in the routine-care paediatric dataset.** Linear mixed-effects models for cortical grey matter, deep grey matter, and lesion volume on a paediatric real-world cohort, stratified by treatment allocation. Brain region volume changes over time in this real-world cohort. Median values are shown as a dot, and the whiskers show the 95% confidence intervals. *N* = 161 patients. 72 patients received high-efficacy treatment, and 89 received moderate efficacy treatment. FLAIR Fluid Attenuated Inversion Recovery, CGM Cortical Grey Matter, DGM Deep Grey Matter. Source data are provided as a Source Data file.

deep grey matter, all three tools reported similar treatment effects. MindGlide estimated a treatment effect of 0.15% (95% CI 0.03−0.27%], *P* = 0.015) for FLAIR images and 0.10% (95% CI −0.01−0.21%], *P* = 0.078) for T2 images, while longitudinal SAMSEG estimated 0.16% (95% CI −0.04−0.36%], *P* = 0.124) for FLAIR images and 0.03% (95% CI −0.07−0.12%], *P* = 0.602) for T2 images, and WMH-Synthseg reported a treatment effect of 0.16% (95% CI 0.03−0.29%], *P* = 0.019) for FLAIR images and 0.14% (95% CI 0.03−0.24%], *P* = 0.010) for T2 images.

WMH-Synthseg performed better than both SAMSEG and longitudinal SAMSEG on in the PPMS dataset. Therefore, we only used WMH-Synthseg as a comparator to MindGlide for our analysis of the routine-care clinical dataset. We visually inspected WMH-Synthseg and MindGlide in our routine-care clinical dataset to assess gross segmentation failures, which included 433 baseline contrasts from 161 patients. WMH-Synthseg demonstrated a significant failure rate, particularly in scans exceeding a thickness of 5 mm. Out of the 433 contrasts that we visually assessed, WMH-Synthseg failed to segment 65 (15%) of the scans, whereas MindGlide exhibited a markedly lower failure rate of only 6 (1%). Of the 6 instances where MindGlide was unable to successfully segment the scans, only one was successfully processed by WMH-Synthseg; the remaining five failed with both methods. Figure 7a shows one example where both, MindGlide and WMH-Synthseg successfully segmented a clinical trial scan. Figure 7b shows a scan from our routine-care clinical dataset with a slice thickness of 7 mm, where WMH-Synthseg's segmentation failed.

### Consistency
**Cross-sectional results.** Figure 8 shows MindGlide segmentation's strong agreement (except for one region) across MRI contrasts (T1, T2, T2-FLAIR, PD) of the same brain in 19 regions. ICC values for brain regions ranged from 0.85 to 0.98, except for the optic chiasm (ICC

0.59). MS lesions demonstrated an ICC of 0.95 (95% CI [0.93, 0.95]) across contrasts. We used our PPMS dataset for this analysis.

In our SPMS dataset (the only cohort with both 2D and 3D T1-weighted acquisitions) we analysed consistency between MindGlide-derived volumes from 3D-T1 scans (1×1×1 mm) vs 2D-T1 scans (1 × 1 × 3 mm). The intraclass correlation coefficients or ICC were 0.929 for lesion, 0.918 for CGM and 0.943 for DGM. We visualised this correlation using a scatter plot in Supplementary Fig. 1.

### Longitudinal: inter-contrast consistency of percentage brain volume change (PBVC)
We evaluated the inter-contrast agreement of longitudinal total brain volume changes. In the PPMS dataset, T1 vs. FLAIR showed an ICC of 0.91 (95% CI [0.88, 0.93]), T1 vs PD had an ICC of 0.916 (95% CI [0.90, 0.93]) and for T1 vs. T2 we calculated an ICC of 0.93 (95% CI [0.91, 0.94]). In the SPMS dataset, the ICC between T1 and T2 was 0.81 (95% CI [0.73, 0.87]). All the images used for this analysis were 2D.

### Longitudinal comparison of 2D and 3D derived volumes
For the deep grey matter in the SPMS dataset (which had both 2D and 3D T1 scans as well as 2D T2 scans), the annual rate of percentage volume loss across both treatment groups was 0.521% [0.346−0.696] for 3D-T1 acquisition, 0.513% [95% CI: 0.308−0.718] for 2D-T1 and 0.474% [95% CI: 0.301−0.645] for 2D-T2 acquisition. The annual rate of percentage volume loss in the cortical grey matter was 0.462% [95% CI: 0.318−0.606] for 3D-T1 acquisition, 0.295% [95% CI: 0.165−0.425] for 2D-T1 and 0.256% [95% CI: 0.139−0.377] for 2D-T2 acquisition. Without ground truth available, we assessed the relative sensitivity loss of the 2D approach compared to the 3D approach. Comparing 3D-T1 and 2D-T1, 2D-T1 showed a 1.54% lower volume loss rate than 3D-T1 for deep grey matter and 36.15% for cortical grey matter. The 2D-T2 acquisition showed a 9.02% lower volume loss rate for deep grey matter than 3D-T1. The sensitivity loss was more pronounced in cortical grey matter, where 2D-T2 detected 44.59% less volume loss than 3D-T1. Comparing 2D-T1 and 2D-T2, 2D-T2 showed a 7.60% lower volume loss rate than 2D-T1 for deep grey matter and 13.22% for cortical grey matter.

## Discussion
Our work establishes the capability to extract multiple clinically relevant MRI biomarkers from a single MRI contrast. MindGlide demonstrates superior performance in multiple key areas compared to state-of-the-art: it more closely aligned with ground truth lesion segmentation, significantly outperformed existing tools in processing routine-care clinical scans (99% success rate vs 85% for WMH-SynthSeg), and showed enhanced sensitivity in detecting cortical grey matter changes, while it performed similarly in deep grey matter segmentation. Furthermore, it captured treatment effects on disease activity (as shown by lesion accrual) and neurodegeneration (as shown by cortical and deep grey matter tissue losses) in clinical trials and routine care hospital settings across a wide age range of trials and hospital settings. This tool significantly streamlines analysis and will enable large-scale research using diverse and often incomplete clinical MRI datasets – an advantage for routine-care studies looking at archival data. Below we will first discuss a comparison of MindGlide with state-of-the-art software (SAMSEG and WMH-Synthseg) looking at segmentation metrics, treatment effects and clinical correlation. We then discuss MindGlide's consistency across MRI contrasts and highlight the differences between clinical trial scans and routine-care scans.

Our results demonstrate that clinically meaningful tissue segmentation and lesion quantification are achievable even with limited MRI data and single contrasts not typically used for these tasks (e.g., T2-weighted MRI without FLAIR). We established the validity and reliability of these findings both cross-sectionally and longitudinally. We compare MindGlide to two state-of-the-art methods: an established contrast-agnostic segmentation method (SAMSEG[4]) and a recently

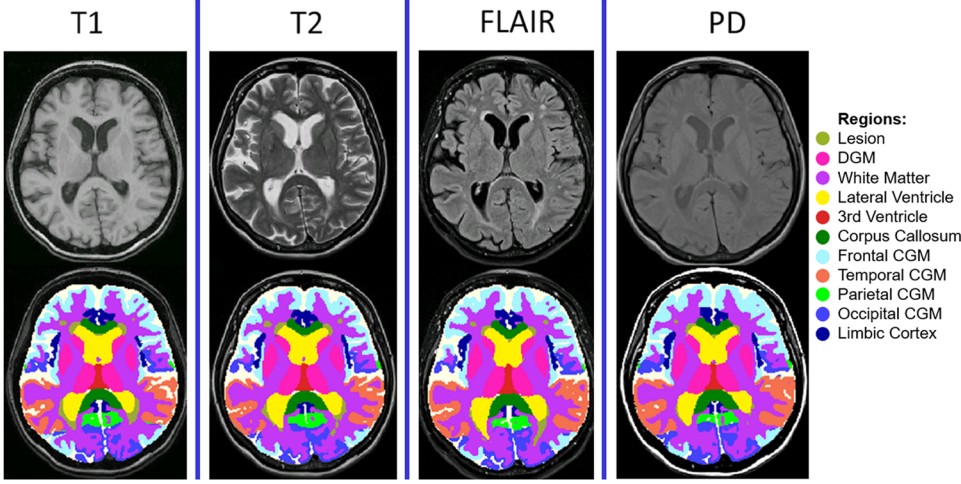

**Fig. 5 | Example segmentations from various contrasts.** The figure shows separate segmentations by the MindGlide model on 2D T1-weighted, T2-weighted, FLAIR and PD contrasts in the PPMS trial. The top row shows the unprocessed ("raw") scans, and the bottom row shows labels or segmentations corresponding to anatomical regions, in addition to white matter hyperintensities (or hypointensities in the case of T1-weighted contrast).

**Table 2 | Treatment effects in the external validation clinical trials across different MRI contrasts**

| Trial | Variable | Contrast[a] | Placebo volume change [95% CI] | Treatment volume change [95% CI] | p-value[b] |
|---|---|---|---|---|---|
| SPMS (simvastatin vs placebo) | Hyperintense lesions | T2 | 1.122 [0.987, 1.256] | 0.768 [0.520, 1.017] | 0.055 |
| | Hypointense lesions | 3D-T1 | 1.875 [1.666, 2.084] | 1.071 [0.686, 1.457] | 0.006 |
| | CGM | T2 | −1.792 [−2.089, −1.495] | −0.704 [−1.254, 0.155] | 0.008 |
| | | 3D-T1 | −2.912 [−3.266, −2.558] | −1.630 [−2.283, −0.976] | 0.009 |
| | DGM | T2 | −0.205 [−0.234, −0.176] | −0.102 [−0.155, −0.048] | 0.009 |
| | | 3D-T1 | −0.234 [−0.263, −0.204] | −0.105 [−0.159, −0.050] | 0.002 |
| PPMS (ocrelizumab vs placebo) | Hyper intense Lesions | T2-FLAIR | 1.042 [0.911, 1.174] | 0.141 [0.053, 0.229] | <0.001 |
| | Hyper intense Lesions | PD | 0.633 [0.535, 0.731] | 0.091 [0.025, 0.157] | <0.001 |
| | Hypointense lesions | T1 | 1.225 [1.112, 1.338] | 0.649 [0.573, 0.725] | <0.001 |
| | Hyper intense Lesions | T2 | 1.104 [0.990, 1.217] | 0.400 [0.324, 0.476] | <0.001 |
| | CGM | T2-FLAIR | −2.342 [−2.722, −1.963] | −1.778 [−2.033, −1.524] | 0.016 |
| | | PD | −2.310 [−2.752, −1.868] | −1.683 [−1.980, −1.386] | 0.021 |
| | | T1 | −2.485 [−2.753, −2.217] | −2.183 [−2.363, −2.002] | 0.066 |
| | | T2 | −2.335 [−2.606−−2.065] | −1.638 [−1.820, −1.457] | <0.001 |
| | DGM | T2-FLAIR | −0.200 [−0.234, −0.167] | −0.156 [−0.178, −0.133] | 0.028 |
| | | PD | −0.220 [−0.255, −0.186] | −0.144 [−0.167, −0.121] | <0.001 |
| | | T1 | −0.212 [−0.245, −0.179] | −0.172 [−0.194, −0.150] | 0.049 |
| | | T2 | −0.172 [−0.203, −0.141] | −0.143 [−0.164, −0.123] | 0.130 |

This table provides details about MindGlide-derived treatment effects in the two clinical trial datasets used in our external validation analysis. A mixed-effects model was used to calculate treatment effects (see Methods, Statistical Analysis).

*95% CI* 95% confidence interval, *CGM* cortical grey matter, *DGM* deep grey matter, *FLAIR* Fluid-Attenuated Inversion Recovery, *PD* proton density, *2D* two dimensional (non-isotropic voxels), *3D* three dimensional (isotropic 1×1×1 mm resolution), *SPMS* secondary progressive multiple sclerosis, *PPMS* primary progressive multiple sclerosis.

[a]All the MRI contrasts were two dimensional or 2D unless specified as 3D (1×1×1 mm). Volume changes are in ml. The real-world cohort is not included here because there was no untreated group (41% received Interferon Beta, 38% Ocrelizumab and the rest other disease modifying treatments).

[b]treatment effect p value, or p value of the difference between volume change rates in treatment vs placebo groups from the mixed effects model.

published segmentation method (WMH-Synthseg[7]) that was based on a previous model called SynthSeg[1–3]. MindGlide performed best in lesion load estimation, dice score and sensitivity (Fig. 2a, b). Treatment effects on lesion volume detected by MindGlide were larger in magnitude than those detected by WMH-Synthseg. When looking at the results of MindGlide and state-of-the-art approaches, MindGlide-derived lesion volumes demonstrated a treatment effect more closely aligned with ground truth values, than SAMSEG, which overestimated, and WMH-Synthseg, which underestimated the treatment effect. While ground truth comparisons were not feasible for regional brain volumes due to the impracticality of manual segmentation of these regions, we were able to compare the relative performance of

SAMSEG, WMH-Synthseg and MindGlide in detecting volumetric changes between treatment groups. MindGlide revealed larger differences in regional brain volume changes between treatment groups compared to the other tools, indicating enhanced sensitivity in detecting subtle changes. Overall, these results support MindGlide as a suitable tool for measuring treatment effects in clinical trials.

At baseline, volumetric measurements extracted using our method showed on average higher correlation with EDSS than SAMSEG or WMH-Synthseg. In our study, WMH-Synthseg, despite having lower dice coefficients compared to SAMSEG, demonstrated a stronger correlation with EDSS. This can be attributed to the relationship between lesion volume and EDSS. WMH-Synthseg tends to estimate

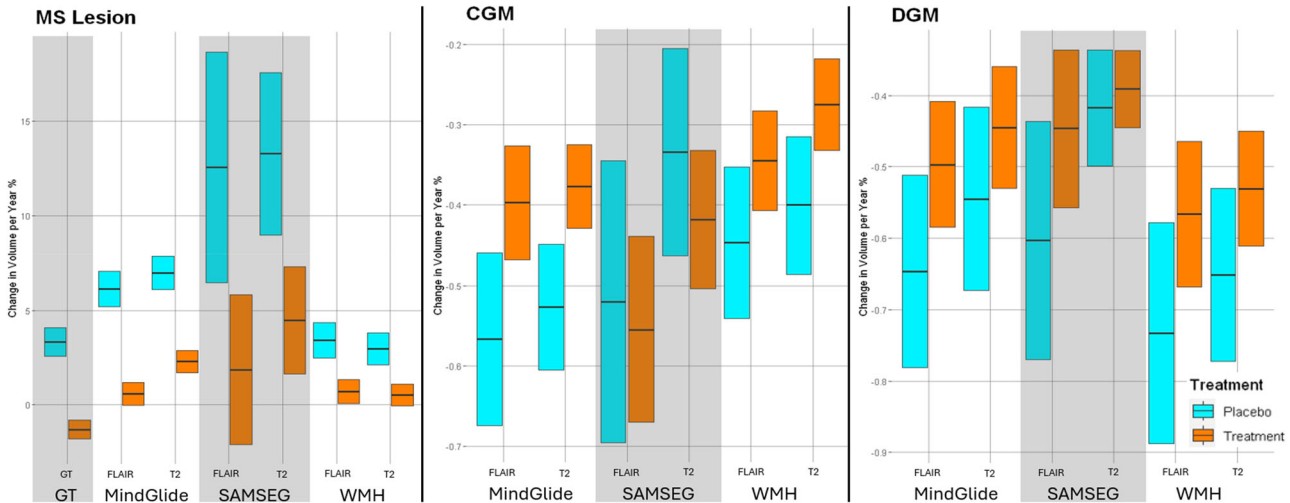

**Fig. 6 | Comparison of longitudinal changes of brain regions and lesion volumes of MindGlide, longitudinal SAMSEG and WMH-Synthseg.** This figure compares derived percentage volume changes per year of MindGlide, longitudinal SAMSEG and WMH-Synthseg for lesion volume, CGM and DGM separated by treatment groups. We used the PPMS clinical trial for this comparison because it was the largest of our datasets and the only one that includes manually segmented lesion volumes by expert neuroradiologists which we used as ground truth. The effect size calculated using MindGlide-derived lesion volume changes is closest to the ground truth. WMH-Synthseg-derived lesion volume change in the placebo group is closest to the ground truth for both FLAIR and T2 images. Ground truth lesion accrual rate was −1.304% per annum in the treatment group and 3.33% per annum in the placebo group. For FLAIR images, MindGlide detected a lesion accrual rate of 0.64% per annum in the treatment group and 5.95% in the placebo group, compared to 1.863% and 12.566% for longitudinal SAMSEG and 0.56% and 3.11% for WMH-Synthseg. With T2 images, MindGlide showed lesion accrual rates of 2.151% and 6.775% for treatment and placebo groups, while longitudinal SAMSEG showed 4.47% and 13.277% and WMH-Synthseg showed 0.359% and 2.813% respectively. The differences between the three tools in measuring CGM and DGM changes are minor compared to lesion volume changes except for the CGM estimates of longitudinal SAMSEG. Here, especially in T2 images, longitudinal SAMSEG estimates more atrophy in the treatment group (−0.418% p.a.) than in the placebo group (−0.334% p.a.), although these differences are not significant ($p = 0.291$) PPMS dataset. $N = 680$. All boxes display medians (centre line in each box) and 95% CI (upper and lower bound of each box). A mixed-effects model was used to calculate treatment effects (see Methods, Statistical Analysis). FLAIR Fluid Attenuated Inversion Recovery, CGM Cortical Grey Matter, DGM Deep Grey Matter, GT Ground Truth. Source data are provided as a Source Data file.

lesion volumes closer to those derived from MindGlide and ground truth than those from SAMSEG. The lower dice score of WMH-Synthseg suggests limited spatial overlap, which is a result of both high false positives and high false negatives, indicating a poor performance in spatial accuracy. Conversely, SAMSEG's higher precision and lower sensitivity led to fewer false positives and a lower dice score. This analysis underscores the limitations of using dice scores as the sole metric for evaluating segmentation tools, which is why we combined various metrics to provide a comprehensive assessment of segmentation performance.

Overall, segmenting different structures was highly consistent except for optic chiasm, which had moderate consistency (ICC of 0.59). This can be explained by the smaller size of the optic chiasm compared to all other MindGlide labels. A single voxel discrepancy within this region wields a proportionately larger impact on the ICC, magnifying the effect of any spatial variations. Furthermore, its close encirclement by cerebrospinal fluid can obscure the chiasm's boundaries in imaging contrasts, reducing the dice score[25]. The intra-class correlation analysis across different MRI contrasts demonstrated the consistency and reliability of percentage brain volume change measurements or PBVC obtained by MindGlide, although ICC across contrasts was higher in the PPMS dataset than in the SPMS dataset.

While our proposed model demonstrates an advantage over state-of-the-art models trained on smaller cohorts of MS patients, it is important to recognise that these improvements may not be as pronounced when analysing clinical trial scans, which often feature controlled conditions and high-quality imaging. However, the benefits of our model become significantly more evident in the context of routine-care scans, where variability in image acquisition and image resolution can pose substantial challenges. In these cases, our model's enhancements lead to dramatic improvements in lesion segmentation and

analysis, thereby offering valuable insights into real-world clinical scenarios.

As expected, there was lost sensitivity in detecting volume changes between two and three-dimensional scans. Comparing 3D-T1 scans with 2D-T2 and 2D-T1 acquisitions revealed a differential impact on sensitivity. In deep grey matter there was only a 2% reduction in detecting atrophy using 2D-T1 and 9% reduction using 2D-T2 acquisitions. In cortical grey matter the reduction in detecting atrophy were 36% using 2D-T1 and 45% using 2D-T2 acquisitions. The comparison between 2D-T1 and 2D-T2 scans showed stability, with 2D-T2 resulting in an 8% lower volume loss rate for deep grey matter and a 13% lower rate for cortical grey matter. These findings will pave the way to incorporate fewer contrasts of the same resolution in MRI protocols, maintaining sensitivity while optimising efficiency, whilst 3D acquisitions are still needed for more detailed analysis.

Our work contributes to the evolution of MRI processing tools that streamline previously time-consuming pipelines. Running the model on consumer-grade graphical processing unit (GPU) hardware took on average, 37 s (see Supplementary Material). Efficiency is especially valuable for MS MRIs, where multimodal imaging is traditionally used to extract biomarkers. The typical workflow has involved intensity inhomogeneity correction[26], followed by automatic segmentation of white matter lesions using T2-FLAIR and three-dimensional T1-weighted MRI. To mitigate the misclassification of hypointense lesions as grey matter (which share a similar intensity profile), anatomical T1-weighted MRIs may undergo lesion filling after affine registration with T2-FLAIR images[27]. Subsequently, hand-labelled T1-weighted MRIs (known as atlases) are non-linearly registered to the patient's T1-weighted MRI[28,29]. Labels from the co-registered atlases are then fused using various fusion algorithms, followed by probabilistic segmentation to differentiate tissue classes (white matter, grey matter, and cerebrospinal fluid)[30]. Moving towards

## a) Trial data

## b) Routine clinical data

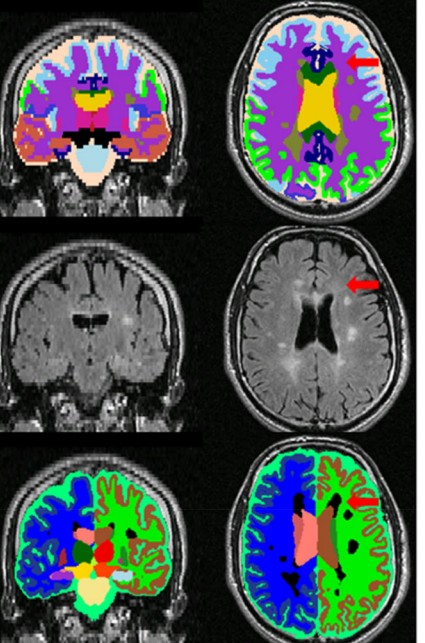
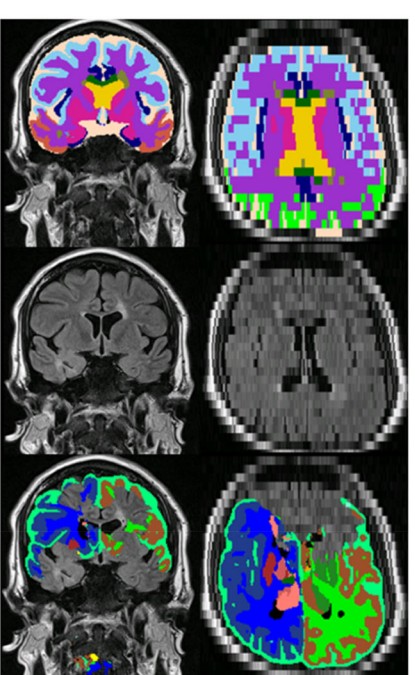

**MindGlide segmentation**

**FLAIR image**

**WMH Synthseg segmentation**

**Fig. 7 | Examples of MindGlide segmentations and WMH-Synthseg segmentation.** Examples of segmentation masks acquired using MindGlide and WMH-Synthseg. **a** Segmentation masks of a scan from our PPMS trial dataset with the segmentation mask acquired using MindGlide on top and the segmentation mask acquired using WMH-Synthseg on the bottom. Areas labelled as lesion are more conservative defined in the MindGlide segmentation mask (olive colour) than in the WMH-Synthseg segmentation mask (black). The red arrow points at an area that is incorrectly defined as lesion by WMH-Synthseg segmentation but not by

MindGlide. **b** Segmentation masks of a scan from our routine clinical dataset (RRMS) with the segmentation mask acquired using MindGlide on top and the segmentation mask acquired using WMH-Synthseg on the bottom. WMH-Synthseg fails to segment an image acquired in anterior-posterior direction with 6 mm thick slices (as seen in the frontal area of the transverse view and multiple areas of the coronal view in (**b**)). Most segmentation tools are designed to use superior-inferior acquisition directions (as in (**a**)), while MindGlide allows segmentation of images acquired in any direction.

newer deep-learning-based pipelines allows fast turnaround times for automatic MRI analysis and real-world implementations.

Our study benefits from a large sample size for model training and external validation. Importantly, our findings generalised across datasets and MRI contrasts. Our training used only FLAIR and T1 images, yet the model successfully processed new contrasts (like PD and T2) from different scanners and periods encountered during external validation. This success is due to the domain randomisation during synthetic data generation (or augmentation), enabling cross-contrast generalisation as has been shown before[1,7]. It is important to note that MindGlide, like WMH-Synthseg, is a 3D convolutional neural network with the same core architecture. The varying performances in segmenting lesions and detecting treatment effects on grey matter structures are due to the diversity of data used to train these models. We used a combination of real and synthetic scans, while previous studies used synthetic or real scans (but not both). The effect of data diversity is well-known in the machine learning community[31]. However, quantifying improvements caused by using various training regimes with real and synthetic data needs further work and was out of the scope of our study.

While we found high ICC values across all segmented brain structures, interpreting the ICC values for MS lesions is complex. We analysed the ICC across different MRI contrasts, including FLAIR, T2, T1, and PD. We analysed the ICC across different MRI sequences during external validation, including FLAIR, T2, T1, and PD. While the MindGlide model was trained on both T1 and FLAIR images, the lesions used for training came from the FLAIR contrast, but the domain randomisation enabled the model to generalise to unseen contrasts. While improving generalisability, this approach blurs pathological specificity

and is a limitation of our contrast-agnostic approach. T1 hypointensities are pathologically distinct from FLAIR hyperintensities[32]. Therefore, while the ICC values provide an essential insight into the tool's reliability, the difference in pathophysiological representation between contrasts necessitates a careful approach to interpreting these results. For example, chronic T1 hypo-intensities, often called "black holes," indicate MS lesions characterised by axonal loss and tissue destruction[33]. The ICC values obtained for MS lesions must, therefore, be considered within the context of these different imaging signatures. A high ICC value might suggest that while the segmentation tool is consistent across contrasts, it may not fully distinguish the complex nature of lesion pathology that varies between T1 and T2/FLAIR contrasts. Nonetheless, as explained above, this approach enables using data to enable a new avenue of research on archival real-world, data.

In our mixed-effects models for estimating treatment effects, we included intracranial volume (ICV) as an extra covariate to account for individual differences at baseline and growth trajectories observed in a paediatric cohort[34–37]. While using the volume-to-ICV ratio as a dependent variable could yield comparable results, our approach maintains the original volume scale, enhancing interpretability and clinical relevance across diverse age groups. The analysis of our routine-care paediatric dataset reveals significant variability in imaging acquisitions, which poses challenges for comparing volume changes across different contrasts. With a heterogeneous range of slice thicknesses and incomplete data for some patients—where only certain contrasts were acquired—our ability to draw definitive conclusions regarding treatment effects is limited. Additionally, as is the case for any observational cohort, treatment effect estimation and causal

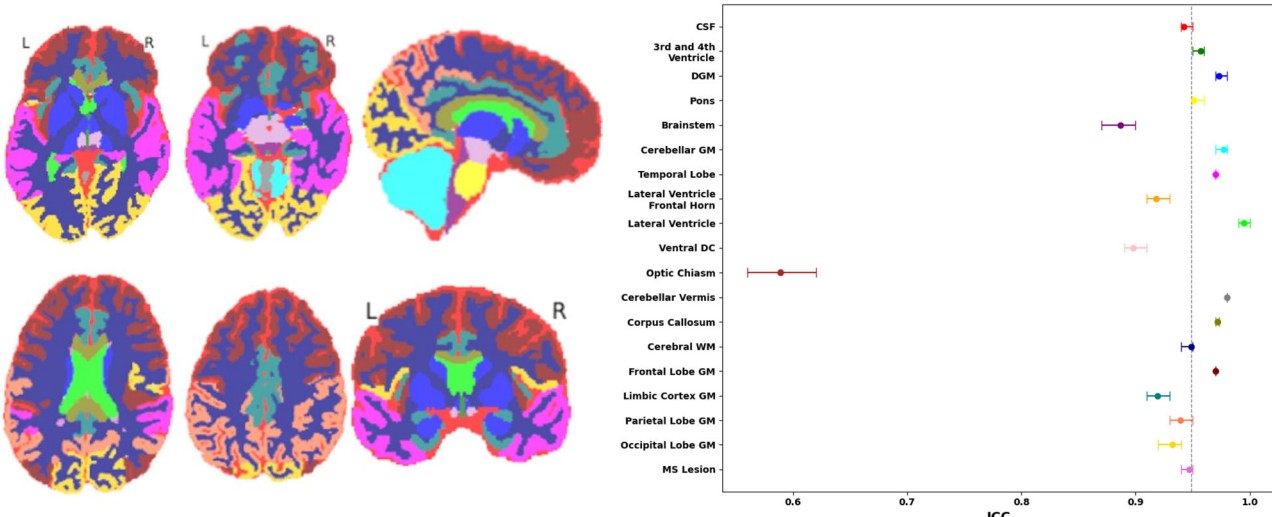

**Fig. 8 | Consistency of regional segmentations across MRI contrasts.** Consistency of segmented regions or labels across multiple MRI contrasts measured by the intraclass Correlation Coefficients (ICCs). On the left, coloured brain maps depict all 19 brain region labels: CSF (Cerebrospinal Fluid), 3<sup>rd</sup> and 4<sup>th</sup> Ventricle, DGM (Deep Grey Matter), Pons, Brainstem, Cerebellar GM (Grey Matter), Temporal Lobe, Lateral Ventricle Frontal Horn, Lateral Ventricle, Ventral DC (Diencephalon), Optic Chiasm, Cerebellar Vermis, Corpus Callosum, Cerebral WM (White Matter), Frontal Lobe GM, Limbic Cortex GM, Parietal Lobe GM, and Occipital Lobe GM,

along with MS (Multiple Sclerosis) Lesions. The right side presents ICC values ranging from 0 to 1 for these regions, providing a quantitative measure of the consistency across multiple MRI contrasts. Higher ICC values indicate greater consistency in the measurement of a particular brain region. Dots represent median intraclass Correlation Coefficient and error bars display 95% confidence intervals. A vertical dashed line marks the median intraclass Correlation Coefficient across all regions. PPMS dataset, baseline images, $N = 699$. Source data are provided as a Source Data file.

conclusions are extremely challenging. For example, we observed an average reduction for patients on moderate-efficacy group (although not statistically significant) using FLAIR images but significant increase in T2-weighted MRI lesions. Despite this wide variability in absolute values, the relative difference between treatment groups was statistically significant and relatively stable across comparisons and brain regions. These results show that MindGlide can enable quantitative insights from highly variable image acquisitions which were previously unanalysable.

Our study has several limitations. The primary limitation is reduced sensitivity in detecting treatment effects using 2D versus 3D scans. While MindGlide showed strong agreement (ICC) across both scan types in the SPMS dataset, the lower resolution of 2D images affects segmentation performance significantly. This can limit detection of small lesions and subtle volume changes, particularly in longitudinal studies tracking tissue changes and volumetric assessments of deep and cortical grey matter. Although 2D scans are more clinically accessible, 3D imaging remains optimal for detailed segmentation. However, our findings suggest that fewer imaging contrasts may suffice for accurate lesion and atrophy detection, potentially reducing scan time and improving resource efficiency in clinical care and research. Additionally, our current implementation is restricted to brain scans. Spinal cord MRI is also widely available in routine care setting and is strongly associated with disability. Future work should expand our approach to the entire central nervous system. We chose the number of labels or segmentations to be 19 by merging smaller labels from the atlas to ensure efficiency and lower computational expense during inference time with a view for implementation within under-resourced research settings (e.g., non-research hospitals). Therefore, current implementation is not intended for detailed segmentations (for example a thalamic volume instead of deep grey matter volume) of brain structures.

In conclusion, we developed and validated a contrast-agnostic deep learning model that can quantify MRI biomarkers from routine care and clinical trial datasets from varying single MRI input contrasts.

## Methods

This study received ethical approval from the Institutional Review Board under the auspices of the International Progressive Multiple Sclerosis Alliance (www.progressivemsalliance.org) at the Montreal Neurological Institute, Canada (IRB00010120) and by the Great Ormond Street Hospital Research and Development Department (reference: 16NC10). Written informed consent was acquired from participants during data acquisition for clinical trials. For routine care MRI scans, consent was waived for processing anonymised data.

### Study design

For training our model we sampled the training data from the International Progressive MS Alliance data repository at the Montreal Neurological Institute. Supplementary Table 1 lists the trials included in the Progressive MS Alliance data repository and Table 1 summarises the patient characteristics of our training and validation datasets. For validation we used 5 different datasets unseen during training. These datasets are from the MS-STAT trial[24], ORATORIO trial[20], a routine-care dataset of paediatric relapsing-remitting MS patients and two open source datasets[38,39].

### Model development

Figure 1 illustrates our training and external validation strategy and the associated steps. We developed MindGlide, using "nnU-Net", a 3D CNN building on the widespread U-Net architecture[40]. nnU-Net yields state-of-the-art results (e.g., it has won several recent challenges[6,41]) while featuring automatic self-configuration, thus bypassing the costly hyperparameter tuning procedure. We trained MindGlide to simultaneously segment brain grey and white matter regions and MS lesions, accommodating real-world MRI variations and artefacts that often hinder traditional image processing software. Our primary goal was to ensure generalisation across MRI contrasts with minimal or no pre-processing at inference, even for contrasts unseen during training (for example, for PD and T2-weighted that were not in our training data). This aligns with successful approaches in other brain imaging studies[1,2].

**Generating training labels.** Supervised models like MindGlide require large, accurately labelled datasets for robust performance. Manually creating such labels, considered the 'gold standard', is time-consuming and impractical for diverse real-world data, especially when the lower quality of scans hinders manual labelling. We used existing segmentations from our datasets of phase two and three clinical trials previously published[42] with Geodesic Information Flows software (GIF v3.0) as explained in our previous publication[43], with additional manual quality control. These labels, derived from the Neuromorphometrics atlas (http://neuromorphometrics.com), were grouped into 18 regions (Supplementary Table 3). To reduce the number of labels and make MindGlide more comparable to other brain image segmentation tools we performed this label grouping according to the hierarchical model of the Mindboggle project (https://mindboggle.info/braincolor). We used a validated lesion segmentation model (a convolutional neural network)[42,44], to generate lesion masks, creating a single file with 20 labels (18 brain regions, one lesion, one background) for training. For feasibility, we employed existing expert-labelled 'ground truth' lesion segmentation data for external validation in fewer individuals, as described below.

**Image pre-processing for model training.** We employed a minimal pre-processing pipeline. We first standardised image resolutions to 1.0 mm isotropic voxels, per the nnU-Net design[6,41]. We then extracted $128 \times 128 \times 64$ voxel patches using a sliding window technique, to optimise memory and computational efficiency during training. While we used data augmentation during training (see below), no further pre-processing was performed at inference.

**Image augmentation and synthetic data.** Data augmentation artificially expands training data diversity through random modification, enhancing model generalizability and mitigating overfitting. To minimise post-training pre-processing and broaden MindGlide's adaptability, we used two techniques: (1) distorting real scans in their geometry and image intensities and (2) generating synthetic ones. Synthetic data generation offers greater flexibility than mere distortion. We employed domain randomisation (Fig. 1a), resulting in intensity variations that prepared the model for diverse MRI contrasts. As shown in Fig. 1d, we performed image augmentation with T1-weighted and FLAIR scans during training.

We used SynthSeg version 2.0 for synthetic data generation and MONAI version 1.2.0 for augmentation during training[1,3,5,45]. We generated synthetic scans of varying contrasts directly from the training dataset's labels (units are as defined by the software)[3]:

Left-right flipping (0.5 probability).
Scaling (uniform distribution, bounds: 0.85–1.15).
Rotation (uniform distribution, bounds: −15–15 degrees).
Elastic deformation (scale: 0.04, standard deviation: 1).
Bias field corruption (scale: 0.25, standard deviation: 0.5).
Random low-resolution resampling (uniform distribution, 1–9 mm per dimension).
Domain randomisation[3] (varying voxel intensities of synthetic scans per tissue class)

Figure 1a illustrates examples of the synthetic data generated. The model architecture, detailed in the Supplementary Material (Model Architecture), has one input channel (receiving a single MR contrast) and 20 output channels (generating 20 labels).

**External validation using independent (unseen in training) datasets**
We performed cross-sectional and longitudinal analyses to assess MindGlide's performance for its validation and reliability in unseen cohorts.

**Cross-sectional analyses**
We used the first available visit in each of the longitudinal datasets for cross sectional analysis. This involved clinical validation by correlating segmented structure and lesion volumes with Expanded Disability Status Scale (EDSS) scores, comparing MindGlide, SAMSEG and WMH-Synthseg results[4,7]. We chose SAMSEG and WMH-Synthseg because they are both recently introduced models, publicly available, as part of Freesurfer, and are among the few models that can segment multiple different contrasts[46,47].

Regarding comparison across MRI contrasts, we used intra-class correlation analysis. Segmentation Consistency across Contrasts: In the PPMS trial, the only dataset with PD, T2, T1, and FLAIR contrasts, we assessed the agreement of segmentations for the same brain structures across these contrasts. We used a hierarchical intraclass correlation coefficient (ICC) to account for the fact that these measurements were taken from the same individuals, which introduces inherent correlation (ICC 3). Only in the SPMS trial, both 3D-T1 and 2D-T1 imaging data was available, and we used these scans to calculate ICC between different resolutions.

**Longitudinal validation in unseen cohorts**
We evaluated MindGlide's ability to detect known treatment effects by analysing data from two successful clinical trials: MS-STAT[24] (placebo vs. simvastatin in secondary progressive MS) and ORATORIO[20] (ocrelizumab vs. placebo in primary progressive MS). Our aim was to demonstrate the capability of MindGlide in detecting known treatment effects using MRI contrasts that have never been used for this purpose (e.g., 2D T2-weighted MRI). We calculated the intra-class correlation coefficient (ICC) for percentage brain volume change with MindGlide segmentations and SIENA algorithm[48] (PBVC), a key trial outcome measure, across MRI contrasts (FLAIR, T2-weighted, T1-weighted, and PD in the PPMS trial and T1 and T2-weighted MRI in the SPMS trial). For longitudinal software comparison, in the PPMS trial, we calculated treatment effects using WMH-Synthseg and MindGlide only without SAMSEG, because WMH-Synthseg showed better performance in cross-sectional comparisons with SAMSEG.

Additionally, we used a routine-care dataset of paediatric relapsing-remitting MS patients from three UK hospitals (Great Ormond Street Hospital, Evelina London Children's Hospital and Birmingham Children's Hospital) to study the longitudinal evolution of lesions and brain structures based on available MRI contrasts (T1-weighted, T2-weighted and FLAIR). We excluded scans from participants whose FLAIR image slice thickness differed by more than a factor of three across follow-up scans. We did not exclude T1-weighted and T2-weighted MRIs because their slice thicknesses varied by less than a factor of three across visits. We categorised patients as receiving high-efficacy (ocrelizumab, natalizumab, rituximab or cladribine) and moderate efficacy treatments (interferon betas, fingolimod, dimethyl fumarate or teriflunomide)[49,50].

**Power analysis**
We performed a power analysis based on MindGlide-derived treatment effects for each contrast to estimate the sample sizes required for a hypothetical clinical trial designed to detect treatment effects using MindGlide on only a single MRI contrast. We used the R pwr library for this analysis.

**Reliability analysis in unseen cohorts**
Lesion Segmentation across Software: We compared lesion segmentations produced by MindGlide, SAMSEG and WMH-Synthseg against ground truth labels (hand-labelled segmentations). We used the same ICC analysis as explained in the cross-sectional analysis above. We measured longitudinal reliability using the ISBI dataset, calculating ICC between raters and MindGlide. See Supplementary Material for details.

## External validation using manual, expert-segmented lesions

Manual lesion segmentation by expert neuroradiologists is considered the gold standard in MS. We used two open-source lesion segmentation datasets (called MS-30 and ISBI[38,39]) and assessed cross-sectional performance against manual lesion segmentations (consensus in MS-30, expert rater in ISBI) using lesion volume and voxel-wise spatial metrics (e.g., Dice score, a standard metric for image segmentation overlap). Cross-sectionally, we assessed lesion load and voxel-wise spatial metrics on 50 FLAIR images from 35 patients, and longitudinally, we calculated the intraclass coefficient (ICC) between raters and MindGlide on the ISBI dataset. For more details, please refer to the Supplementary Material.

## Statistical analysis

We used R version 4.3.0 for all analyses. In cross-sectional analysis, we assessed correlations between segmented brain volumes and EDSS using Spearman's rank correlation and Fisher Z scores because EDSS is an ordinal variable.

We used linear mixed-effects models to estimate treatment effects. Each regional volume or lesion load was the dependent variable in a separate model. Fixed independent variables included time, treatment group, their interaction (time x treatment group), and intracerebral volume (ICV). Random effects, nested by visit within participant ID, accounted for repeated measures and within-participant variability. We did not adjust for other variables because the comparisons were made in data from randomised controlled trials in treatment and control arms. In the real-world data we did not adjust for age (all participants were in their adolescence) and used the same fixed independent variables and random effects. In real-world paediatric dataset, we did not perform a head-to-head comparison of moderate versus high efficacy treatment because participants were not randomised and the small number of children with MS did not allow for causal modelling.

We performed ICC with the Pengouin statistical package for Python 3. We used ICC3 because we had a fixed set of "raters" (segmentation of the same structures by different software [MindGlide vs SAMSEG vs WMH-Synthseg] or from different contrasts).

## Reporting summary

Further information on research design is available in the Nature Portfolio Reporting Summary linked to this article.

# Data availability

Data are controlled by pharmaceutical companies and are proprietary. Requests for the sole purpose of reproducing the results of the study can be made available upon on contacting the corresponding author which will endeavour to make it available within a month of submitting a request Source data are provided with this paper.

# Code availability

The code, trained models, and computational environment (container) for MindGlide are publicly accessible at https://github.com/MS-PINPOINT/mindGlide (https://doi.org/10.5281/zenodo.14725884)[51].

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

## Acknowledgements

This study was funded by the National Institute for Health and Care Research (NIHR) Advanced Fellowship Round 7 to Dr Arman Eshaghi (NIHR302495). O.C. was supported by the NIHR Research Professorship (RP-2017-08- ST2-004). The views expressed are those of the author(s) and not necessarily those of the NIHR or the Department of Health and Social Care. A.J.T., O.C. and D.C. were supported by the National Institute for Health and Care Research University College London Hospitals Biomedical Research Centre. The clinical trial data collection was partly supported by an award from the International Progressive Multiple Sclerosis Alliance (award reference number PA-1412-02420). J.E.I. has received funding from awards NIH 1RF1AG080371, 1R21NS138995, 1RF1MH123195, 1R01AG070988, 1R01EB031114, and 1UM1MH130981.

## Author contributions

Conceptualisation: P.G., A.E., J.S.; Experiments: P.G., J.W., A.E., J.S.; Neuroimage data collection: O.A., S.D.G.R., T.R., E.W., J.C., D.A., D.L.C., Y.H., A.E.; Drafting and review: P.G., J.W., A.E.; Writing – review and editing: P.G., J.W., O.A., B.B.V., J.S., S.D.G.R., O.K., T.R., E.W., J.C., D.A., D.L.C., C.H., S.N., D.C., J.E.I., F.B., G.J.M.P., N.P.O., Y.H., A.T., D.C.A., O.C., A.E.; Supervision: J.W., J.S., A.T., D.C.A., O.C., A.E., Approval of the draft: all authors.

## Competing interests

D.C. is a consultant for Hoffmann-La Roche. In the last three years he has been a consultant for Biogen, has received research funding from Hoffmann-La Roche, the International Progressive MS Alliance, the MS Society, the Medical Research Council, and the National Institute for Health Research (NIHR) University College London Hospitals (UCLH) Biomedical Research Centre, and a speaker's honorarium from Novartis. He co-supervises a clinical fellowship at the National Hospital for Neurology and Neurosurgery, London, which is supported by Merck. F.B. acts as a member of the steering committee or Data Safety Monitoring Board for Biogen, Merck, ATRI/ACTC and Prothena. Consultant for Roche, Celltrion, Rewind Therapeutics, Merck, IXICO, Jansen, Combinostics. Research agreements with Merck, Biogen, GE Healthcare, Roche. Co-founder and shareholder of Queen Square Analytics LTD. O.C. is a NIHR Research Professor (RP-2017-08-ST2-004); over the last 2 years, member of independent DSMB for Novartis; she gave a teaching talk in a Merck local symposium, and contributed to an Advisory Board for Biogen; she is Deputy Editor of Neurology, for which she receives an honorarium; she has received research grant support from the MS Society of Great Britain and Northern Ireland, the NIHR UCLH Biomedical Research Centre, the Rosetree Trust, the National MS Society, and the NIHR-HTA. C.H. reports grant support from the MRC and MS Society. She has served as a consultant to Novartis, Roche, UCB and Sanofi. S.N. has received research funding from the Canadian Institutes of Health Research, the International Progressive MS Alliance, the Myelin Repair Foundation, Immunotec, and F. Hoffman LaRoche, not related to the current work; he is a consultant for Sana Biotech, has received a speaker's honorarium from Novartis Canada, and is a part-time employee of NeuroRx Research. In the last 3 years, J.C. has received support from the Health Technology Assessment (HTA) Programme (National Institute for Health Research, NIHR), the UK MS Society, the US

National MS Society and the Rosetrees Trust. He is supported in part by the NIHR University College London Hospitals (UCLH) Biomedical Research Centre, London, UK. He has been a local principal investigator for a trial in MS funded by MS Canada. A local principal investigator for commercial trials funded by: Ionis and Roche; and has taken part in advisory boards/consultancy for Biogen, Contineum Therapeutics, InnoCare, Lucid, Merck, NervGen, Novartis and Roche. G.J.M.P. is a shareholder and director of, and receives salary from, Bioxydyn Limited. He is a shareholder and director of Queen Square Analytics Limited. He is a shareholder and director of Quantitative Imaging Limited. D.C.A. is a shareholder and director of Queen Square Analytics Limited. In the past three years, A.E. has received research grants from the Medical Research Council (MRC), NHS England, Imperial College Healthcare Trust, National Institute for Health and Social Care Research (NIHR), Innovate UK, Biogen, Merck, and Roche. He has served as an advisory board member of Merck Serono and Bristol Myers Squib. He is the founder and equity stakeholder in Queen Square Analytics Limited. He serves on the editorial board of Neurology (American Academy of Neurology). The remaining authors declare no competing interests.

## Additional information

[1]Queen Square Multiple Sclerosis Centre, Department of Neuroinflammation, Faculty of Brain Sciences, University College London, London, UK. [2]UCL Hawkes Institute, University College London, London, UK. [3]Centre for Advanced Research Computing (ARC), University College London, London, UK. [4]Department of Paediatric Neurology, Guy's and St Thomas' NHS Foundation Trust, London, United Kingdom. [5]Birmingham Children's Hospital, Birmingham, United Kingdom. [6]Institute of Health and Neurodevelopment, Aston University, Birmingham, UK. [7]McConnell Brain Imaging Centre, Montreal Neurological Institute-Hospital, Department of Neurology and Neurosurgery , McGill University, Montreal, Quebec, Canada. [8]Department of Paediatric Neurology, Great Ormond Street Hospital, London, UK; Institute of Neurology, UCL, London, UK. [9]National Institute for Health Research University College London Hospitals Biomedical Research Centre (BRC), London, UK. [10]Martinos Center for Biomedical Imaging, Massachusetts General Hospital, Harvard Medical School, Charlestown, MA, USA. [11]Computer Science and Artificial Intelligence Laboratory, Massachusetts Institute of Technology, Cambridge, MA, USA. [12]Department of Radiology and Nuclear Medicine, Amsterdam UMC, Vrije Universiteit, Amsterdam, The Netherlands. [13]UCL Department of Medical Physics and Biomedical Engineering, University College London, London, UK. [14]Bioxydyn Limited, Manchester, UK. [15]UCL Department of Computer Science, University College London, London, UK. ✉e-mail: p.goebl@ucl.ac.uk

