## [Transparent Peer Review file · Nature Communications]

Repurposing Clinical MRI Archives for Multiple Sclerosis Research with a Flexible, Single-Contrast Approach: New Insights from Old Scans

Corresponding Author: Dr Philipp Goebel

Version 0:

Reviewer comments:

Reviewer #1

(Remarks to the Author)

The work is important as it might have a significant impact on the field. While further studies are necessary to confirm the findings, the large dataset used for training and the external validation sets are reassuring.

The methodology employed is a well-established AI-based procedure for image segmentation and has been successfully applied here to old MRI scans of people with multiple sclerosis.

I have no major criticisms regarding the methods and results. However, I am not sure how this work on old datasets can lead to "shorter MRI acquisition protocols and reduced costs for MS trials" in the future. All data reported here were from old MRI scans with 2D acquisition, and I am not convinced that the method would apply to the current clinical 3D MRI acquisition.

In general, the manuscript would benefit from a limitations section in the discussion.

(Remarks on code availability)

Reviewer #2

(Remarks to the Author)

In this paper, the authors propose to use a convolutional neural network to automatically quantify both lesions and neuroanatomy in MS patients from a single MRI contrast. By training the network on both synthetic data and real data acquired with a plethora of scanners and MRI contrasts, the method is able to handle single-contrast MRIs across a variety of sequences and scanners -- similar to the synthseg method it seems to be based upon. Experiments are conducted on images obtained from two clinical trials and one routine-care dataset in MS. It is concluded that the proposed method opens the door to quantifying MS disease progression from cheaper, lower-quality acquisitions, and that it outperforms two other methods (synthseg and samseg from freesurfer) on this task.

Although it is clear that the authors have developed a well-working method, I have major concerns about the scientific quality of the manuscript, both in terms of how well the experiments support some of the claims that are made, and about the overall quality of presentation:

A. The abstract and introduction section contain very strong claims about the potential of the proposed method to remove the need for acquiring high-quality multi-contrast MRI data, going as far as claiming that the method "paves the way for shorter MRI acquisition protocols and reduced costs for MS trials" and "potentially [makes] clinical trials less costly by reducing the need for multi-contrast acquisitions". However, while it is shown that some disease effects can, indeed, be detected even from low-resolution, single-contrast T2-weighted images, an analysis of how much sensitivity is lost compared to the standard setting where a high-resolution T1-weighted image is used to quantify atrophy, is missing from all experiments. In those few instances where this information can inadvertently be gleaned from the provided plots (e.g., cortical gray matter and PBVC for the SPMS dataset in fig 3), the results are often dramatic: atrophy rates are underestimated by a factor of two

when 3mm T2 is used instead of 1mm T1, and ICC values of total brain volume plummet from around 0.92 to around 0.73. If the authors really want to maintain that it may be cost-effective to drop high-resolution T1-weighted scans, then they should evaluate how the increased number of subjects that need to be enrolled in a trial (power analysis) is offset by a reduction in MRI scanning costs.

B. More generally, there is a tendency in the paper to only highlight findings that align with the preferred narrative, whereas less convenient results are not discussed and are often not clearly presented in the provided plots and tables:

- one example is the issue of high-resolution T1s to measure cortical atrophy already mentioned before: apart from the issue already identified in fig 3, which is not discussed in the paper, also fig 4(left) seems to show the same effect, but we can't be sure because the image resolutions for this dataset is never clarified in the paper. In fig 2, the comparison of cortical gray matter quantification is only between T2 and FLAIR, whereas T1 is absent and we are never told which dataset(s) were used to generate this figure (presumably a mix of PPMS with 3mm T1s and SPMS with 1mm T1s?). The same issue arises again in fig 5 and 8 (and suppl. figs 1 and 2): high visual correspondences and ICC values are reported across input MRI contrasts, but these results are only for 3mm T1s in the PPMS dataset and don't show the corresponding results for 1mm T1s in the SPMS dataset, which would surely be a lot worse. (We are actually never told which dataset is used to generate fig 8, but based on the low ICC for total brain volume in SPMS reported in fig 3(d), it must be PPMS.) In the manuscript, the effect of image resolution is never discussed, sometimes going as far as stating the opposite of what is visible in the figures (see for instance the strong wording on pages 18 and 22 and in the figure caption about the ICC results shown in fig 3(d)).

- another example of a less-pleasing result that is never discussed is MS lesions segmented from T2 instead of FLAIR: fig 3(a) and fig 6(left) seem to show a threefold difference in volume change detection between those two contrasts in the treatment group, whereas fig 4(right) shows an increase in the moderate efficacy group for one contrast but a decrease in the other (in addition to a twofold difference in volume change in the high efficacy group). I think the reasons and/or implications of such discrepancies should at least be discussed.

C. While the authors demonstrate that their method can detect disease effects from low-resolution single-contrast MRI scans, they also claim that it does so better than the existing tools synthseg and samseg (stating in the abstract that it "uniquely enables quantitative analysis"). In as far as this is true for lesion quantification (although the much higher specificity of samseg, and the fact that this can be traded off to increase sensitivity in the software, is not discussed), it is not clear that is also the case for atrophy measurements:

- the only direct comparison between the methods' capability to quantify atrophy is in fig 2(c), where gray matter volume is correlated with EDSS. But for deep gray matter there doesn't seem to be a statistically significant difference with synthseg (although the text says, confusingly, it is with samseg), and for cortical gray matter there are no statistically significant differences altogether (except for the strange samseg result for T2, which the authors should verify is not a sign error). Nevertheless, the discussion section contains strong wording, in three different places, about how the proposed method "outperforms" synthseg and samseg in this respect.

- none of the other experiments that investigate how the proposed method quantifies atrophy also report the performance of synthseg and samseg, which is a missed change to investigate the strengths and weaknesses of the three different tools. I find this is especially problematic for the longitudinal experiments -- ultimately the presumed target application -- since e.g., samseg can explicitly exploit the longitudinal dimension to further increase sensitivity.

- again, there seems to be a tendency to cherry-pick results. For instance, fig 6 highlights a lack of sensitivity in synthseg in measuring lesion volume in one particular dataset (PPMS), but doesn't repeat the experiment for the twin dataset (SPMS); doesn't analyze or discuss the atrophy measures that are also shown for both methods; and doesn't comment on the threefold discrepancy between FLAIR and T2 for the proposed method's lesion change estimations in the treatment group (synthseg provides much more consistent estimates). Similarly, fig 7 highlights two specific subjects in which synthseg fails but the proposed method succeeds, but without statistics on a larger number of subjects is unclear how representative this figure is -- presumably some subjects could also be found where synthseg succeeds and the proposed method fails?

D. In its current form, the manuscript is very difficult to follow because it lacks structure and doesn't always provide the information needed to correctly interpret the results:

- the section entitled "External validation using independent (unseen in training) datasets" (pages 10-12) should be carefully restructured. It fails to describe the cross-sectional experiments where ICCs are computed when the MRI contrast is changed; it fails to mention how exactly cross-sectional experiments are performed on data this is longitudinal; it describes (twice!) an experiment whose results are only given in the suppl. material; and it describes three times (!) the same experiment of comparing automatic with manual lesion segmentations, each time providing additional information.

- relevant information regarding the datasets that are used is incomplete and often provided piecemeal in different parts of the manuscript. For instance, patient and imaging characteristics are only provided in the results section, and even then important information (such as the image resolution of the various MRI contrasts in the routine-care dataset, or the number of follow-up scans in all three datasets) is missing.

- as already mentioned before, it is never clarified which datasets fig 2(c) and fig 8 are based upon. Further sources of

unnecessary confusion include: not mentioning that the upper part of fig 3(a-c) is about PPMS and the lower part about SPMS; not mentioning which datasets were used for the PBVC values of fig 3(d) or how it was computed (the text even mentions SIENA?), and only defining PBVC in a figure caption; and variously changing the direction of the y-axis and the order of results between/across subplots of fig 3 and fig 4.

(Remarks on code availability)

Reviewer #3

(Remarks to the Author)

Goebel et al. propose a deep learning model to extract robust MRI biomarkers from single-contrast scans. The model is validated on data from several clinical trials and an observational cohort. The availability of such a large and diverse dataset offers the opportunity to validate the performance of previously published state-of-the-art models which had not been tested on such heterogeneous data, and to rigorously compare them against newly developed tools. Despite the promising setting, I have some concerns (listed below) on the model evaluation methods adopted, and more generally on the actual novelty of the proposed tool against existing state-of-the-art algorithms. The model architecture, augmentation techniques, and training strategy were leveraged from previous literature. Despite using a larger dataset for training, the proposed model seems to only bring a moderate advantage w.r.t. state-of-the-art models that were trained on substantially smaller cohorts of MS patients – except for a few aspects. Dice scores against manual lesion segmentation, for instance, did show a remarkable improvement from state-of-the-art (although on a small validation set), but this information is somewhat submerged by several other less meaningful results which are maybe given too much emphasis.

Major points:

1) My main concern is on the claims of the authors when it comes to treatment effect estimation. Estimating a larger effect does not necessarily mean getting closer to the true effect – in fact, the effect may be overestimated. Both MindGlide and state-of-the-art models should be compared against “ground truth”, i.e., the treatment effect as estimated using hand-labelled lesions. Was manual lesion segmentation available for at least a subset of the longitudinal data used in the study? If not, unless a previously estimated “ground truth” treatment effect is available as a target value for validation, I doubt that any meaningful conclusion can be drawn when it comes to the ability of the proposed model to correctly estimate treatment effect. The same concern applies for treatment effects on brain atrophy.

2) I think the sentence “MindGlide-derived lesion load had a stronger correlation to EDSS than the state-of-the-art” is a bit misleading and should be softened, as the difference between correlation coefficients is non-significant in 3 out of 4 comparisons.

On a related note: it is reasonable to assume that a better lesion segmentation leads to stronger correlation between lesion load and EDSS. If this is correct, the correlation with EDSS should be even stronger for ground truth hand-labelled lesions – which are taken as the gold standard for lesion segmentation. Can this be checked from the available data? The “ground truth” correlation would also provide a target value for the correlation with EDSS, to be compared against the one obtained with MindGlide and other software.

3) In the proposed mixed-effects models for estimating treatment effect, the region (or lesion) volume is modelled as a linear function of time, with group-specific slopes. In this context, it does not seem correct to me to include ICV as an additive covariate with a global coefficient which is the same across all subjects. Was there a specific hypothesis motivating this choice? I think it would be more correct to directly use the ratio volume/ICV as a dependent variable, so that the group-specific time slope captures the variation in normalised volume, which is comparable across subjects.

Minor points:

1) Computing time is reported for MindGlide but not for existing models.

2) Was inter-contrast consistency evaluated also for state-of-the-art models? This would be another interesting metric for comparing MindGlide against existing models – especially since robust biomarker extraction from any MRI contrast is one of the main motivations reported by the authors for the development of this new model.

3) The WMH-Synthseg algorithm is trained using only synthetic images generated from segmentation labels. In the Synthseg paper, a supervised version of the model, trained using real scans, had been tested. While performing better than Synthseg when cross-validated in the same dataset, it proved unable to generalise the segmentation when tested on an external dataset. MindGlide is trained on a dataset including both real scans, and synthetic scans generated with SynthSeg. Have you checked whether the inclusion of the real scans actually leads to any advantage in model performance as compared to training only on synthetic scans (as is the case for SynthSeg)?

(Remarks on code availability)

I did not run the code, but it looks clean and well commented. The git repo does provide a readme file with detailed instructions on installation and usage.

Version 1:

Reviewer comments:

Reviewer #1

(Remarks to the Author)

None

(Remarks on code availability)

Reviewer #2

(Remarks to the Author)

I wish to congratulate the authors on both their work and on the way they responded to the many additional reviewer requests. I have no further comments.

(Remarks on code availability)

Reviewer #3

(Remarks to the Author)

The authors addressed most of my comments. I only have one remark left about treatment effect. I think the authors did a great job determining ground-truth treatment effect on lesion volume. However, my former comment also mentioned treatment effect on atrophy, which still lacks validation against ground truth. Larger differences in brain volumes do not necessarily correspond to distances closer to reality. This is particularly relevant considering that SAMSEG did overestimate treatment effect on lesion accrual: a similar situation may arise with atrophy. If it is not feasible to obtain ground-truth treatment effect on atrophy, this limitation should be clearly stated in the text.

Two very minor suggestions to make the reading easier:

- In the section "Treatment effects on lesion accrual", they authors may want to specify more clearly that the lesion volumes they refer to are MindGlide-derived.
- In Figure 2, it would be best to keep the same colour for GT across panels.

(Remarks on code availability)

Response to Reviewer Comments

Reviewer #1 (Remarks to the Author):

Comment 1: The work is important as it might have a significant impact on the field. While further studies are necessary to confirm the findings, the large dataset used for training and the external validation sets are reassuring. The methodology employed is a well-established AI-based procedure for image segmentation and has been successfully applied here to old MRI scans of people with multiple sclerosis.

Response: We thank the reviewer for their time and comments on our manuscript.

Comment 2: I have no major criticisms regarding the methods and results. However, I am not sure how this work on old datasets can lead to "shorter MRI acquisition protocols and reduced costs for MS trials" in the future. All data reported here were from old MRI scans with 2D acquisition, and I am not convinced that the method would apply to the current clinical 3D MRI acquisition.

Response: We agree with the reviewer on the statements on acquisition cost and have adjusted them in the revision as shown below.

Abstract

"MindGlide uniquely enables quantitative analysis of archival single-contrast MRIs, unlocking insights from untapped hospital datasets."

Regarding acquisition types, we apologise for the unclearness of the types of data used. We included 3D MRI acquisitions and have now clarified as shown below. We added additional analysis to our manuscript to compare how our model performs on 3D and 2D acquisitions. The results show excellent agreement using the Intraclass Correlation Coefficient (ICC) between 2D and 3D acquisitions. These results demonstrate that our model can perform equally well on both 2D and 3D acquisitions.

Methods, Page 17, line 405

"Only in the SPMS trial, both 3D-T1 and 2D-T1 imaging data was available, and we used these scans to calculate ICC between different resolutions."

Results, Page 13, Line 300

*“In our SPMS dataset (the only cohort with both 2D and 3D T1-weighted acquisitions) we analysed consistency between MindGlide-derived volumes from 3D-T1 scans (1x1x1mm) vs 2D-T1 scans (1x1x3mm). The intraclass correlation coefficients or ICC were 0.929 for lesion, 0.918 for CGM and 0.943 for DGM. We visualised this correlation using a scatter plot in **Supplemental Figure 1.**”*

Results, Page 9, Line 186

“Using 3D T1-weighted MRI, the rate of hypointense lesion accrual was significantly faster in the placebo group than in the treatment group (1.874 mL/year vs 1.071 mL/year; $P=0.005$).”

Results, Page 10, Line 208 (regarding the use of 3D T1 scans for the cortical grey matter)

“This effect was consistent across both T2-weighted (-0.704 mL/year (95% CI [-1.254 - 0.155]) vs. -1.792 mL/year (95% CI [-2.089 - -1.495]), $P = 0.008$) and 3D T1-weighted MRI (-1.630 mL/year (95% CI [-2.283 - -0.976]) vs. -2.912 mL/year (95% CI [-3.266 - -2.558]), $P = 0.009$).”

Results, Page 10, Line 211 (regarding the use of 3D T1 scans for the deep grey matter)

“The DGM had a slower rate of loss in the treatment group compared to placebo for both ... and 3D T1-weighted contrasts (0.105 ml/year (95% CI [-0.159 - -0.050]) vs. 0.234 ml/year (95% CI [-0.263 - -0.204]), $P = 0.001$).”

Comment 4: In general, the manuscript would benefit from a limitations section in the discussion.

Response: we have added a limitation section as below:

Discussion, Page 26, line 618

“Our study has several limitations. The primary limitation is reduced sensitivity in detecting treatment effects using 2D versus 3D scans. While MindGlide showed strong agreement (ICC) across both scan types in the SPMS dataset, the lower resolution of 2D images affects segmentation performance significantly. This can limit detection of small lesions and subtle volume changes, particularly in longitudinal studies tracking tissue changes and volumetric assessments of deep and cortical grey matter. Although 2D scans are more clinically accessible, 3D imaging remains optimal for detailed segmentation. However, our findings suggest that fewer imaging contrasts may suffice for accurate lesion and atrophy detection, potentially reducing scan time and improving resource efficiency in clinical care and research. Additionally, our current

implementation is restricted to brain scans. Spinal cord MRI is also widely available in routine care setting and is strongly associated with disability. Future work should expand our approach to the entire central nervous system. We chose the number of labels or segmentations to be 19 by merging smaller labels from the atlas to ensure efficiency and lower computational expense during inference time with a view for implementation within under-resourced research settings (e.g., non-research hospitals). Therefore, current implementation is not intended for detailed segmentations (for example a thalamic volume instead of deep grey matter volume) of brain structures. “

Reviewer #2 (Remarks to the Author):

Comment 1: In this paper, the authors propose to use a convolutional neural network to automatically quantify both lesions and neuroanatomy in MS patients from a single MRI contrast. By training the network on both synthetic data and real data acquired with a plethora of scanners and MRI contrasts, the method is able to handle single-contrast MRIs across a variety of sequences and scanners -- similar to the synthseg method it seems to be based upon. Experiments are conducted on images obtained from two clinical trials and one routine-care dataset in MS. It is concluded that the proposed method opens the door to quantifying MS disease progression from cheaper, lower-quality acquisitions, and that it outperforms two other methods (synthseg and samseg from freesurfer) on this task.

Response: We thank the reviewer for their comments on our manuscript.

Comment 2: Although it is clear that the authors have developed a well-working method, I have major concerns about the scientific quality of the manuscript, both in terms of how well the experiments support some of the claims that are made, and about the overall quality of presentation:

Response: We have now included more detailed analyses to address these important comments about experiments. Specifically, we have analysed how MindGlide performs across different MRI resolutions and contrasts by comparing volumetric consistency and sensitivity in both high-resolution 3D and lower resolution 2D acquisitions. Our intra class correlation analysis (comment 3, reviewer #2) demonstrates that MindGlide achieves high concordance between lesion and grey matter volumes derived from both acquisition types. We have also revised the manuscript's presentation as specified in our responses to comments below.

Comment 3: The abstract and introduction section contain very strong claims about the potential of the proposed method to remove the need for acquiring high-quality multi-contrast MRI data, going as far as claiming that the method "paves the way for shorter MRI acquisition protocols and reduced costs for MS trials" and "potentially [makes] clinical trials less costly by reducing the need for multi-contrast acquisitions". However, while it is shown that some disease effects can, indeed, be detected even from low-resolution, single-contrast T2-weighted images, an analysis of how much sensitivity is lost compared to the standard setting where a high-resolution T1-weighted image is used to quantify atrophy, is missing from all experiments.

Response: Concerning claims on reducing clinical trial costs, we agree and have adjusted the statements to better reflect our results as shown below.

We added additional analysis for the loss of sensitivity when high resolution scans are used. In our revisions, we compared MindGlide-derived volumes of high-resolution 3D-T1 scans (1x1x1mm) and lower-resolution 2D-T1 volumes (1x1x3mm) (see also

Comment 2 from Reviewer 1) with our SPMS dataset (which had both 2D and 3D data available). We performed an intraclass correlation coefficient (ICC) analysis of MindGlide-derived volumes of hypointense lesions, cortical and deep grey matter (CGM and DGM) between 3D and 2D MRI acquisitions. The correlation coefficients were 0.929 for lesion, 0.918 for CGM and 0.943 for DGM. When looking at the treatment effect on brain atrophy in cortical and deep grey matter, as expected and as the reviewer rightly mentions, sensitivity was lost: for deep grey matter, the annual rate of volume loss across both treatment groups was 0.521% [0.346 - 0.696] for 3D-T1 acquisition and 0.474% [95% CI: 0.301 - 0.645] for 2D-T2 acquisition. The annual rate of volume loss in cortical grey matter was 0.462% [95% CI: 0.318 - 0.606] for 3D-T1 acquisition and 0.256% [95% CI: 0.139 - 0.377] for 2D-T2 acquisition. Without ground truth available, we assessed the relative sensitivity loss of the 2D approach compared to the 3D approach. Comparing 3D-T1 and 2D-T1, 2D-T1 showed a 1.54% lower volume loss rate than 3D-T1 for deep grey matter and 36.15% for cortical grey matter. The 2D-T2 acquisition showed a 9.02% lower volume loss rate for deep grey matter than 3D-T1. The sensitivity loss was more pronounced in cortical grey matter, where 2D-T2 detected 44.59% less volume loss than 3D-T1. Comparing 2D-T1 and 2D-T2, 2D-T2 showed a 7.60% lower volume loss rate than 2D-T1 for deep grey matter and 13.22% for cortical grey matter.

We have made the following changes:

Abstract (we have removed the claims on the costs of clinical trials)

“MindGlide uniquely enables quantitative analysis of archival single-contrast MRIs, unlocking insights from untapped hospital datasets.”

Results, Page 13, Line 300

“In our SPMS dataset (the only cohort with both 2D and 3D T1-weighted acquisitions) we analysed consistency between MindGlide-derived volumes from 3D-T1 scans (1x1x1mm) vs 2D-T1 scans (1x1x3mm). The intraclass correlation coefficients or ICC were 0.929 for lesion, 0.918 for CGM and 0.943 for DGM. We visualised this correlation using a scatter plot in Supplemental Figure 1.”

Results, Page 14, Line 313

“Longitudinal comparison of 2D and 3D derived volumes

For the deep grey matter in the SPMS dataset (which had both 2D and 3D T1 scans as well as 2D T2 scans), the annual rate of percentage volume loss across both treatment groups was 0.521% [0.346 - 0.696] for 3D-T1 acquisition, 0.513% [95% CI: 0.308 - 0.718] for 2D-T1 and 0.474% [95% CI: 0.301 - 0.645] for 2D-T2 acquisition. The annual rate of percentage volume loss in the cortical grey matter was 0.462% [95% CI: 0.318 - 0.606] for 3D-T1 acquisition, 0.295% [95% CI: 0.165 - 0.425] for 2D-T1 and 0.256% [95% CI: 0.139 - 0.377] for 2D-T2 acquisition. Without ground truth available, we assessed the relative sensitivity loss of the 2D approach compared to

the 3D approach. Comparing 3D-T1 and 2D-T1, 2D-T1 showed a 1.54% lower volume loss rate than 3D-T1 for deep grey matter and 36.15% for cortical grey matter. The 2D-T2 acquisition showed a 9.02% lower volume loss rate for deep grey matter than 3D-T1. The sensitivity loss was more pronounced in cortical grey matter, where 2D-T2 detected 44.59% less volume loss than 3D-T1. Comparing 2D-T1 and 2D-T2, 2D-T2 showed a 7.60% lower volume loss rate than 2D-T1 for deep grey matter and 13.22% for cortical grey matter.”

Discussion, Page 23, Line 546

“As expected, there was lost sensitivity in detecting volume changes between two and three-dimensional scans. Comparing 3D-T1 scans with 2D-T2 and 2D-T1 acquisitions revealed a differential impact on sensitivity. In deep grey matter there was only a 2% reduction in detecting atrophy using 2D-T1 and 9% reduction using 2D-T2 acquisitions. In cortical grey matter the reduction in detecting atrophy were 36% using 2D-T1 and 45% using 2D-T2 acquisitions. The comparison between 2D-T1 and 2D-T2 scans showed stability, with 2D-T2 resulting in an 8% lower volume loss rate for deep grey matter and a 13% lower rate for cortical grey matter. These findings will pave the way to incorporate fewer contrasts of the same resolution in MRI protocols, maintaining sensitivity while optimising efficiency, whilst 3D acquisitions are still needed for more detailed analysis.”

Comment 4: In those few instances where this information can inadvertently be gleaned from the provided plots (e.g., cortical gray matter and PBVC for the SPMS dataset in fig 3), the results are often dramatic: atrophy rates are underestimated by a factor of two when 3mm T2 is used instead of 1mm T1, and ICC values of total brain volume plummet from around 0.92 to around 0.73. If the authors really want to maintain that it may be cost-effective to drop high-resolution T1-weighted scans, then they should evaluate how the increased number of subjects that need to be enrolled in a trial (power analysis) is offset by a reduction in MRI scanning costs.

Response: Our assertion of cost reduction potential is primarily based on minimising required MRI sequences across 2D acquisitions rather than obviating the need for 3D acquisitions. Our decision was informed by the current phase 3 MS clinical trials that almost exclusively use 2D acquisitions for feasibility across countries and scanners. For example, in ORATORIO (a phase 3 clinical trial in PP MS) all MRI sequences (T1, T2, T2-FLAIR, PD) were acquired at 2D acquisitions (1x1x3mm) without 3D acquisitions. The Reviewer’s comment is important and therefore we have added power analysis using various MRI sequences with 2D and 3D acquisitions. We have added the results to supplemental table 5.

Results, Page 13, Line 294

“Cross-sectional results

Figure 8 shows MindGlide segmentation’s strong agreement (except for one region) across MRI contrasts (T1, T2, T2-FLAIR, PD) of the same brain in 19 regions. ICC values for brain regions ranged from 0.85 to 0.98, except for the optic chiasm (ICC 0.59). MS lesions demonstrated an ICC of 0.95 (95% CI [0.93, 0.95]) across contrasts. We used our PPMS dataset for this analysis.

In our SPMS dataset we analysed consistency between MindGlide-derived volumes from 3D-T1 scans (1x1x1mm) vs 2D-T1 scans (1x1x3mm). The intraclass correlation coefficients were 0.929 for lesion, 0.918 for CGM and 0.943 for DGM. We visualised this correlation using a scatter plot in Supplemental Figure 1.”

Methods, Page 18, Line 432

“Power analysis

We performed a power analysis based on MindGlide-derived treatment effects for each contrast to estimate the sample sizes required for a hypothetical clinical trial designed to detect treatment effects using MindGlide on only a single MRI contrast. We used the R pwr library for this analysis.”

Supplemental Table 5: Power analysis based on treatment effects derived from MindGlide’s segmentation volumes using a single contrast

Dataset for treatment effect estimation (PPMS = 699 subjects, SPMS = 141 subjects)	Resolution	Contrast	Region	Participants required for 80% power
PPMS	2D	T2	Lesion	76
PPMS	2D	FLAIR	Lesion	62
PPMS	2D	PD	Lesion	94
PPMS	2D	T1	CGM	2166
PPMS	2D	T2	CGM	420
PPMS	2D	FLAIR	CGM	1244
PPMS	2D	PD	CGM	1375
PPMS	2D	T1	DGM	1882
PPMS	2D	T2	DGM	3180
PPMS	2D	FLAIR	DGM	1506
PPMS	2D	PD	DGM	562
SPMS	2D	T2	Lesion	228
SPMS	2D	T1	CGM	496

SPMS	2D	T2	CGM	120
SPMS	3D	T1	CGM	128
SPMS	2D	T1	DGM	592
SPMS	2D	T2	DGM	126
SPMS	3D	T1	DGM	88

Discussion, Page 26, Line 623

“...Although 2D scans are more clinically accessible, 3D imaging remains optimal for detailed segmentation. However, our findings suggest that fewer imaging contrasts may suffice for accurate lesion and atrophy detection, potentially reducing scan time and improving resource efficiency in clinical care and research.”

Comment 5: B. More generally, there is a tendency in the paper to only highlight findings that align with the preferred narrative, whereas less convenient results are not discussed and are often not clearly presented in the provided plots and tables:

Response: Thank you for bringing this to our attention. We addressed each specific concern in the comments below.

Comment 6: - one example is the issue of high-resolution T1s to measure cortical atrophy already mentioned before: apart from the issue already identified in fig 3, which is not discussed in the paper, also fig 4(left) seems to show the same effect, but we can't be sure because the image resolutions for this dataset is never clarified in the paper.

Response: Figure 4 illustrates longitudinal volume changes derived by MindGlide in our real-world cohort, which features substantial variability in image resolution. In this hospital dataset, slice thickness is highly variable, ranging from 0.4 mm to 8.5 mm with considerable differences across contrasts. Specifically, FLAIR and T1 contrasts have a median slice thickness of 1.0 mm, with minimum values of 0.43 mm and 0.45 mm, and maximum values of 7.20 mm and 7.65 mm, respectively. T2-weighted images, however, exhibit a wider distribution, with a median slice thickness of 4.8 mm, a minimum of 0.98 mm, and a maximum of 8.50 mm. We appreciate the reviewer's insight and agree that these differences in spatial resolution are relevant and therefore we have added the image resolution for the real-world dataset to Table 1. Also, Supplemental Figure 4 now shows a distribution of slice thickness in this dataset, separated by contrast.

Supplemental Figure 4: Real-World Dataset: Distribution of Slice Thickness by Contrast

Caption: This histogram shows the frequency of slice thickness (in mm) across the three MRI contrasts in our real-world dataset: FLAIR (red), T1-weighted (green), and T2-weighted (blue). FLAIR and T1 contrasts both have a median slice thickness of 1.0 mm, with minimum values of 0.43 mm and 0.45 mm, respectively, and maximum values of 7.20 mm and 7.65 mm, respectively. T2-weighted images exhibit a broader distribution, with a median slice thickness of 4.8 mm, a minimum of 0.98 mm, and a maximum of 8.50 mm. The distribution highlights a predominance of thinner slices (0-1 mm) in FLAIR and T1-weighted images, while T2-weighted images show peaks around thicker slices.

Comment 7: In fig 2, the comparison of cortical gray matter quantification is only between T2 and FLAIR, whereas T1 is absent and we are never told which dataset(s) were used to generate this figure (presumably a mix of PPMS with 3mm T1s and SPMS with 1mm T1s?).

Response: We agree and have conducted additional analyses incorporating T1-weighted scans (1x1x3 mm) to calculate correlation coefficients between regional brain volumes and EDSS scores. Additionally, we have added the dataset used for each analysis (see below). Results indicated comparable performance across MindGlide, SAMSEG and WMH-Synthseg in the PPMS clinical trial T1 scans. Specifically, for cortical gray matter (CGM), the correlation coefficients were -0.14 (95% CI: -0.21 to -0.06) for MindGlide, -0.11 (95% CI: -0.19 to -0.04) for SAMSEG, and -0.13 (95% CI: -0.20 to -0.05) for WMH-Synthseg. For deep gray matter (DGM), the correlation coefficients were -0.13 (95% CI: -0.20 to -0.06) for MindGlide, -0.11 (95% CI: -0.18 to -0.03) for SAMSEG, and -0.11 (95% CI: -0.19 to -0.04) for WMH-Synthseg. While MindGlide demonstrated the highest correlation between brain volume and EDSS for both CGM and DGM, these differences were not statistically significant (CGM: FLAIR $P = 0.816$, T1 $P = 0.923$; DGM: FLAIR $P = 0.439$, T2 $P = 0.680$, T1 $P = 0.885$) except for CGM T2 where MindGlide and WMH-Synthseg showed a negative correlation while SAMSEG showed a positive correlation between brain volume and EDSS ($P = 0.002$).

In addition to the correlation between lesion volumes and EDSS, we now also added a ground truth using manually labelled lesions by expert neuroradiologists. The calculated correlation coefficient between this ground truth and EDSS was 0.13 (95% CI: 0.06 to 0.20).

For Figure 2c, we used the baseline scans from the PPMS dataset ($N = 699$), and we have clarified this in the figure's caption.

Results, Page 7, Line 141

“As Figure 2c shows, in our PPMS dataset, MindGlide-derived lesion load had a numerically higher correlation with the Expanded Disability Status Scale (EDSS) than the state-of-the-art. However, the differences were not statistically significant for all comparisons. When using T2-FLAIR scans, the correlation between MindGlide-derived lesion load and the EDSS was on average higher (correlation coefficient = 0.127, $P < 0.001$) than the correlations observed with SAMSEG (correlation coefficient = 0.009, $P = 0.813$) or WMH-Synthseg (correlation coefficient = 0.105, $P = 0.005$). When using T2 scans, MindGlide showed significant correlation between lesion load and EDSS (correlation coefficient = 0.150, $P < 0.001$), which was also the case for SAMSEG (correlation coefficient = 0.086, $P = 0.022$) and WMH-Synthseg (correlation coefficient = 0.140, $P < 0.001$). In head-to-head comparisons, we observed a significant difference between the correlation coefficients of MindGlide and SAMSEG lesion loads from FLAIR ($P = 0.026$), but not for T2 ($P = 0.227$). There was no difference between MindGlide and WMH-Synthseg lesion loads for either contrast (FLAIR: $P = 0.680$; T2: $P = 0.850$). As a reference, the correlation coefficient between ground truth expert-labelled hyperintense T2 lesion volumes and EDSS was 0.131 (95% CI: 0.057 - 0.203; $P < 0.001$).

Regarding the correlation between deep grey matter (DGM) volumes and EDSS scores, MindGlide-derived DGM volume demonstrated a correlation coefficient of -0.130 on FLAIR contrasts (95% CI: -0.202 to -0.057; $P < 0.001$), -0.128 on T2-weighted (95% CI: -0.200 to -0.054; $P < 0.001$) and -0.131 on 2D T1-weighted images (95% CI: -0.203 to -0.057; $P < 0.001$). Conversely, SAMSEG-derived DGM volume yielded correlation coefficients of -0.057 (95% CI: -0.130 to 0.018; $P = 0.134$), -0.084 (95% CI: -0.157 to -0.010; $P = 0.026$) and -0.106 (95% CI: -0.178 to -0.031; $P = 0.005$) for FLAIR, T2-weighted and 2D T1-weighted contrasts, respectively. WMH-Synthseg-derived DGM volume yielded correlation coefficients of -0.106 (95% CI: -0.179 to -0.032; $P = 0.005$), -0.112 (95% CI: -0.185 to -0.038; $P = 0.003$) and -0.112 (95% CI: -0.188 - -0.035; $P = 0.004$) for FLAIR, T2-weighted and T1-weighted contrasts. There were no statistically significant differences across the correlation coefficients obtained using MindGlide, SAMSEG and WMH-Synthseg on either FLAIR ($P = 0.439$), T2-weighted ($P = 0.680$) or T1-weighted contrasts ($P = 0.885$).

In the cortical grey matter (CGM), MindGlide-derived volumes and EDSS correlation coefficients were -0.123 on FLAIR (95% CI: -0.195 to -0.049; $P = 0.001$), -0.102 on T2 (95% CI: -0.175 to -0.028; $P = 0.007$) and -0.135 on T1 contrasts (95% CI: -0.207 to -0.061; $P < 0.001$). SAMSEG-derived CGM volume and EDSS correlation coefficients were -0.121 on FLAIR (95% CI: -0.193 to -0.047; $P = 0.001$), 0.053 on T2 (95% CI: -0.021 to 0.126; $P = 0.160$) and -0.114 on T1 contrasts (95% CI: -0.186 to -0.039; $P = 0.002$). WMH-Synthseg-derived CGM volume yielded correlation coefficients of -0.091 (95% CI: -0.164 to -0.017; $P = 0.016$), -0.114 (95% CI: -0.186 to -0.040; $P = 0.026$) and -0.127 (95% CI -0.202 - -0.050; $P = 0.001$) for FLAIR, T2-weighted and T1-weighted contrasts. There was a statistically significant difference between the correlation coefficients when using MindGlide or SAMSEG on T2 contrasts ($P < 0.001$) but not on FLAIR ($P = 0.966$) or T1 ($P = 0.691$). There was no difference between correlation coefficients obtained using MindGlide and WMH-Synthseg (FLAIR: $P = 0.549$; T2: $P = 0.828$; T1: $P = 0.879$).”

Revised Figure 2. Performance comparisons with state-of-the-art and ground truth

*Caption: (A) Boxplot displaying lesion load estimates (mm³) and distributions measured using ground truth manual delineations (grey), MindGlide (blue) and Freesurfer's SAMSEG (orange). Lesion load estimates between Ground truth and SAMSEG and MindGlide and SAMSEG methods were significantly different (paired *t*-tests). (B) Boxplot displaying Dice scores, Sensitivity and Precision measurements for both MindGlide (blue) and SAMSEG (orange) delineated lesions. *** $P < 0.001$, ** $P < 0.01$, * $P < 0.05$, $n=50$. In (C) we calculated Spearman's correlation coefficients for regional brain volumes obtained from MindGlide and Freesurfer's SAMSEG and WMH-Synthseg against the expanded disability status scale (EDSS). The analysis evaluates correlations of lesion, deep grey matter (DGM), and cortical grey matter (CGM) volumes with EDSS, across FLAIR and T2 MRI contrasts. As a ground truth comparator for the correlation between lesion volume and EDSS we used manually*

labelled lesions by expert neuroradiologists. For all tested regions and contrasts MindGlide's output shows on average higher correlations with EDSS scores except for CGM in T2 (although as shown, they are not statistically significantly different across software). Error bars represent 95% CI. For (A) and (B) we used two openly available lesion segmentation datasets as ground truth comparators (N = 50, see supplemental methods). For (C) we used the baseline images of our PPMS dataset (N = 699). GT = ground truth (manually labelled lesion segmentation by expert neuroradiologists)

Comment 8: The same issue arises again in fig 5 and 8 (and suppl. figs 1 and 2): high visual correspondences and ICC values are reported across input MRI contrasts, but these results are only for 3mm T1s in the PPMS dataset and don't show the corresponding results for 1mm T1s in the SPMS dataset, which would surely be a lot worse. (We are actually never told which dataset is used to generate fig 8, but based on the low ICC for total brain volume in SPMS reported in fig 3(d), it must be PPMS.)

Response: We revised the captions of all our figures to specify in detail what dataset we used for each plot and subplot. Also, we added supplemental figure 3 to visualise correlation of MindGlide-derived volumes between 3D-T1 and 2D-T1 scans in our SPMS dataset. The ICC for hypointense lesion was 0.929, for CGM ICC was 0.918 and for DGM ICC was 0.943.

Below we list all revised figures and captions with detailed explanations of the datasets we used.

1. Revised Figure 2: See comment 7 by Reviewer #2.
2. Revised Figure 3

Caption: MindGlide uniquely enables quantifying treatment effects using single MRI contrasts, including those that have never been used for this purpose (e.g., T2-weighted MRI). (A) – (C) shows longitudinal volume changes with results for our PPMS dataset on the top and results for our SPMS dataset on the bottom. (A) illustrates the

annual per cent change in lesion volume detected by MindGlide across FLAIR, PD, T1, and T2 contrasts (resolution: 1x1x3mm) for primary progressive MS (PPMS) and secondary progressive MS (SPMS) cohorts, stratified by treatment allocation. Treatment groups had a reduction in lesion volume accrual compared to placebo across all contrasts. Boxes display medians and 95% CI. (B) depicts the annualised rate of cortical grey matter (CGM) atrophy. MindGlide successfully differentiated between treatment and placebo groups, demonstrating reduced cortical atrophy across all MRI contrasts in treated patients. This is also the case for atrophy rates in deep grey matter (DGM) as seen in (C). There are no FLAIR and PD contrasts available for the SPMS cohort. (D) shows inter-contrast consistency for percentage brain volume changes (PBVC): High intra-class correlation coefficients (ICC) for percent brain volume change (PBVC) across different MRI contrasts within the PPMS dataset (2D), indicating high inter-contrast consistency. This underscores the segmentation tool's robustness and consistency in detecting neurodegenerative changes across various imaging contrasts. In the SPMS dataset we compared PBVC of 2D-T1 images and 2D-T2 images with an ICC-coefficient of 0.81 [95% CI 0.73 - 0.87]. PPMS: N = 680, SPMS: N= 130.

3. Revised Figure 6

Caption: This figure compares derived percentage volume changes per year of MindGlide, longitudinal SAMSEG and WMH-Synthseg for lesion volume, CGM and DGM separated by treatment groups. We used the PPMS clinical trial for this comparison because it was the largest of our datasets and the only one that includes manually segmented lesion volumes by expert neuroradiologists which we used as ground truth. The effect size calculated using MindGlide-derived lesion volume changes is closest to the ground truth. WMH-Synthseg-derived lesion volume change in the placebo group is closest to the ground truth for both FLAIR and T2 images. Ground truth lesion accrual rate was -1.304% per annum in the treatment group and 3.33% per annum in the placebo group. For FLAIR images, MindGlide detected a lesion accrual rate of 0.64% per annum in the treatment group and 5.95% in the placebo group, compared to 1.863% and 12.566% for longitudinal SAMSEG and 0.56% and 3.11% for WMH-Synthseg. With T2 images, MindGlide showed lesion accrual rates of 2.151% and 6.775% for treatment and placebo groups, while

longitudinal SAMSEG showed 4.47% and 13.277% and WMH-Synthseg showed 0.359% and 2.813% respectively. The differences between the three tools in measuring CGM and DGM changes are minor compared to lesion volume changes except for the CGM estimates of longitudinal SAMSEG. Here, especially in T2 images, longitudinal SAMSEG estimates more atrophy in the treatment group (-0.418% p.a.) than in the placebo group (-0.334% p.a.), although these differences are not significant ($p= 0.291$) PPMS dataset. N = 680.

4. New Supplemental Figure 3. Correlation of segmentations between 3D and 2D acquisitions.

Caption: Correlation of volumes derived from 3D-T1 (1x1x1mm) and 2D-T1 (1x1x3mm) MRI sequences for lesions, cortical grey matter (CGM), and deep grey matter (DGM) visualised using scatter plots. MindGlide demonstrates consistent volume estimation across varying image resolutions. The thicker red line represents the regression line, and the thinner red line represents the line of identity. The outlier (red circle) in the CGM plot was caused by a cropped brain in the 3D-T1 scan, resulting in incomplete segmentation. Data source: SPMS dataset (N=141)

Results, Page 13, Line 300

“In our SPMS dataset (the only cohort with both 2D and 3D T1-weighted acquisitions) we analysed consistency between MindGlide-derived volumes from 3D-T1 scans (1x1x1mm) vs 2D-T1 scans (1x1x3mm). The intraclass correlation coefficients or ICC were 0.929 for lesion, 0.918 for CGM and 0.943 for DGM. We visualised this correlation using a scatter plot in Supplemental Figure 1.”

Comment 9: In the manuscript, the effect of image resolution is never discussed, sometimes going as far as stating the opposite of what is visible in the figures (see for instance the strong wording on pages 18 and 22 and in the figure caption about the ICC results shown in fig 3(d)).

Response: We have added additional analyses between image resolutions (2D, and 3D T1-weighted MRI acquisitions) as shown in response to Comments 4 and 8 from Reviewer #2 and Comment 2 from Reviewer #1.

We revised our previous wording and highlighted the high intraclass correlation between different contrasts in our PPMS dataset compared to a lower ICC between 2D-T1 and 2D-T2 scans in our SPMS dataset (which was the only dataset with both 2D and 3D acquisitions):

Results, Page 12, Line 262:

“In the PPMS dataset, T1 vs. FLAIR showed an ICC of 0.91 (95% CI [0.88, 0.93]), T1 vs PD had an ICC of 0.916 (95% CI [0.90, 0.93]) and for T1 vs. T2 we calculated an ICC of 0.93 (95% CI [0.91, 0.94]). In the SPMS dataset, the ICC between T1 and T2 was 0.81 (95% CI [0.73, 0.87]). All the images used for this analysis are 2D. We also updated the visualisation of these results in figure 3d.”

Discussion, Page 22, Line 534:

“The intra-class correlation analysis across different MRI contrasts demonstrated the consistency and reliability of PBVC measurements obtained by MindGlide, although ICC across contrasts was higher in the PPMS dataset than in the SPMS dataset.”

Discussion, Page 26, Line 618

“The primary limitation is reduced sensitivity in detecting treatment effects using 2D versus 3D scans. While MindGlide showed strong agreement (ICC) across both scan types in the SPMS dataset, lower resolution of 2D images affects segmentation performance significantly. This can limit detection of small lesions and subtle volume changes, particularly in longitudinal studies tracking tissue changes and volumetric assessments of deep and cortical grey matter. Although 2D scans are more clinically accessible, 3D imaging remains optimal for detailed segmentation. However, our findings suggest that fewer imaging contrasts may suffice for accurate lesion and atrophy detection, potentially reducing scan time and improving resource efficiency in clinical care and research.”

Comment 10: - another example of a less-pleasing result that is never discussed is MS lesions segmented from T2 instead of FLAIR: fig 3(a) and fig 6(left) seem to show a threefold difference in volume change detection between those two contrasts in the treatment group, whereas fig 4(right) shows an increase in the moderate efficacy group for one contrast but a decrease in the other (in addition to a twofold difference in volume change in the high efficacy group). I think the reasons and/or implications of such discrepancies should at least be discussed.

Response: We agree that in our PPMS dataset (fig 3(a) and fig 6 (left), MindGlide-derived lesion volumes vary between contrasts when looking at specific treatment

groups. Our results show that the treatment effect (difference between control and treatment) is highly consistent across contrasts, although absolute values in each group, as the reviewer rightly mentions, and as expected, vary. For example, T2 and FLAIR-derived lesion volumes show only a small difference in lesion growth per year of around 5% between the treatment effects assessed by each contrast.

In our routine-care paediatric dataset (illustrated in Supplemental Figure 4), we had significant heterogeneity in image acquisitions, typical of hospital settings. Specifically, T1 and T2-FLAIR images have a median slice thickness of 1 mm (ranging from 0.43 mm to 7.65 mm), while T2 images have a median slice thickness of 4.8 mm (ranging from 0.98 mm to 8.5 mm). Furthermore, it is important to note that not all contrasts were acquired for every subject. Some subjects only had T1 images, while others had only T2-FLAIR images. This variability, coupled with the frequent missing data points typical of routine-care datasets, limits our ability to make reliable comparisons of volume changes across sequences in this dataset.

In Figure 4 shows the ability to detect treatment differences across various contrasts despite high heterogeneity, rather than agreement across contrasts. We appreciate the reviewer's insightful comments and added these limitations and their implications to our discussion section.

Discussion, Page 25, Line 605:

"The analysis of our routine-care paediatric dataset reveals significant variability in imaging acquisitions, which poses challenges for comparing volume changes across different contrasts. With a heterogeneous range of slice thicknesses and incomplete data for some subjects—where only certain contrasts were acquired—our ability to draw definitive conclusions regarding treatment effects is limited. Additionally, as is the case for any observational cohort, treatment effect estimation and causal conclusions are extremely challenging. For example, we observed an average reduction for patients on moderate-efficacy group (although not statistically significant) using FLAIR images but significant increase in T2-weighted MRI lesions. Despite this wide variability in absolute values, the relative difference between treatment groups was statistically significant and relatively stable across comparisons and brain regions. These results show that MindGlide can enable quantitative insights from highly variable image acquisitions which were previously unanalysable."

Comment 11: C. While the authors demonstrate that their method can detect disease effects from low-resolution single-contrast MRI scans, they also claim that it does so better than the existing tools synthseg and samseg (stating in the abstract that it "uniquely enables quantitative analysis"). In as far as this is true for lesion quantification (although the much higher specificity of samseg, and the fact that this can be traded off to increase sensitivity in the software, is not discussed), it is not clear that is also the case for atrophy measurements:

Response: We have new analyses to address these excellent points for both lesions and atrophy measurements as follows:

- a) **Comparison with the ground truth:** we added ground truth lesion volumes (independently done from our study and manually corrected by expert neuroradiologists) to a new Figure 6 and compared it with the percentage lesion volume changes derived from MindGlide, longitudinal SAMSEG and WMH-Synthseg. When analysing treatment effects we saw a similar treatment effect in MindGlide-derived lesion volumes (5.31% difference between treatment groups in FLAIR images, 4.62% in T2 images) compared to the ground truth (4.63% difference between treatment groups). Longitudinal SAMSEG overestimated the treatment effect on lesion volumes (10.70% difference between treatment groups in FLAIR images, 8.81% in T2 images) and WMH-Synthseg underestimated the treatment effect (2.71% difference between treatment groups in FLAIR images, 2.44% in T2 images). Additionally, SAMSEG-long shows a broader spread in lesion changes in both T2 and FLAIR images compared to the ground truth and MindGlide-derived volumes, suggesting a higher precision of MindGlide.
- b) **Direct comparisons to state-of-the-art approaches:** MindGlide-derived regional brain volumes showed larger differences between treatment groups than derived brain volumes from longitudinal SAMSEG or WMH-Synthseg. MindGlide-derived CGM volume changes showed a 0.17% difference between treatment groups for FLAIR images and 0.15% for T2 images. Longitudinal SAMSEG-derived CGM volume changes showed a 0.04% difference between treatment groups for FLAIR images and 0.08% for T2 images. WMH-Synthseg-derived CGM volume changes showed a 0.10% difference between treatment groups for FLAIR images and 0.13% for T2 images. For deep grey matter all three tools showed a similar treatment effect. MindGlide estimated a treatment effect of 0.15% for FLAIR images and 0.10% for T2 images, longitudinal SAMSEG estimated a treatment effect of 0.16% for FLAIR images and 0.03% for T2 images and WMH-Synthseg estimated a treatment effect of 0.17% for FLAIR images and 0.12% for T2 images.

Revised Figure 6:

See comment 8 by reviewer #2.

Results, Page 11, Line 248

“Comparing Treatment Effects with MindGlide against Other Segmentation Tools and Ground Truth Lesions in the PPMS clinical trial

In our analysis, MindGlide-derived lesion volumes demonstrated a treatment effect of 5.31% (95% CI [4.50% - 6.12%], $P < 0.001$) difference between treatment groups in

FLAIR images and 4.62% (95% CI [3.88% - 5.37%], $P < 0.001$) in T2 images (Figure 6). This closely aligns with ground truth values, which indicated a 4.63% (95% CI [3.73% - 5.54%], $P < 0.001$) difference between treatment groups. In contrast, longitudinal SAMSEG overestimated the treatment effect, showing a 10.70% (95% CI [3.42% - 17.98%], $P = 0.004$) difference for FLAIR images and 8.81% (95% CI [3.64% - 13.97%], $P = 0.001$) for T2 images, while WMH-Synthseg underestimated the effect with only a 2.56% (95% CI [1.64% - 3.47%], $P < 0.001$) difference in FLAIR images and 2.45% (95% CI [1.69% - 3.22%], $P < 0.001$) in T2 images. Additionally, longitudinal SAMSEG exhibited a broader spread in lesion changes across both T2 and FLAIR images compared to MindGlide-derived volumes, suggesting a higher precision in MindGlide's lesion volume estimation.

Direct Comparisons of Tissue Volumes and Visual Inspection in the PPMS trial and real-world cohort

As Figure 6 shows, in assessing regional brain volumes, MindGlide-derived measurements demonstrated greater treatment effects between treatment groups compared to those obtained from longitudinal SAMSEG or WMH-Synthseg. Specifically, MindGlide-derived CGM volume changes revealed a 0.14% (95% CI 0.04% - 0.24%), $P = 0.006$) difference between treatment groups for FLAIR images and a 0.16% (95% CI 0.09% - 0.23%), $P < 0.001$) difference for T2 images. In comparison, longitudinal SAMSEG-derived CGM volume changes indicated a 0.04% (95% CI - 0.18% - 0.24%), $P = 0.744$) difference for FLAIR images and 0.08% (95% CI -0.07% - 0.24%), $P = 0.288$) for T2 images, while WMH-Synthseg-derived CGM volume changes exhibited a 0.11% (95% CI 0.02% - 0.19%), $P = 0.014$) difference for FLAIR images and a 0.12% (95% CI 0.05% - 0.2%), $P = 0.002$) difference for T2 images. For deep grey matter, all three tools reported similar treatment effects. MindGlide estimated a treatment effect of 0.15% (95% CI 0.03% - 0.27%), $P = 0.015$) for FLAIR images and 0.10% (95% CI -0.01% - 0.21%), $P = 0.078$) for T2 images, while longitudinal SAMSEG estimated 0.16% (95% CI -0.04% - 0.36%), $P = 0.124$) for FLAIR images and 0.03% (95% CI -0.07% - 0.12%), $P = 0.602$) for T2 images, and WMH-Synthseg reported a treatment effect of 0.16% (95% CI 0.03% - 0.29%), $P = 0.019$) for FLAIR images and 0.14% (95% CI 0.03% - 0.24%), $P = 0.010$) for T2 images.”

Comment 12: - the only direct comparison between the methods' capability to quantify atrophy is in fig 2(c), where gray matter volume is correlated with EDSS. But for deep gray matter there doesn't seem to be a statistically significant difference with synthseg (although the text says, confusingly, it is with samseg), and for cortical gray matter there are no statistically significant differences altogether (except for the strange samseg result for T2, which the authors should verify is not a sign error). Nevertheless, the discussion section contains strong wording, in three different places, about how the proposed method "outperforms" synthseg and samseg in this respect.

Response: We agree with the reviewer's statement that there is no significant difference in DGM volume correlation and EDSS between the three tools (Figure 2c).

Our new analysis where we compared treatment effects measured by MindGlide, longitudinal SAMSEG and WMH-Synthseg supports this statement (Figure 6). In contrast, when analysing treatment effects on lesion volume, MindGlide is closer to the ground truth than longitudinal SAMSEG and WMH-Synthseg. We rephrased our wording regarding EDSS correlation and treatment effect estimations of MindGlide, longitudinal SAMSEG and WMH-Synthseg. For this we refer to Comments 7 and 11 by Reviewer #2.

Discussion, Page 21, Line 487

“MindGlide demonstrates superior performance in multiple key areas compared to state-of-the-art: it more closely aligned with ground truth lesion segmentation, significantly outperformed existing tools in processing routine-care clinical scans (99% success rate vs 85% for WMH-SynthSeg), and showed enhanced sensitivity in detecting cortical grey matter changes, while it performed similarly in deep grey matter segmentation.”

Comment 13: - none of the other experiments that investigate how the proposed method quantifies atrophy also report the performance of synthseg and samseg, which is a missed change to investigate the strengths and weaknesses of the three different tools. I find this is especially problematic for the longitudinal experiments -- ultimately the presumed target application -- since e.g., samseg can explicitly exploit the longitudinal dimension to further increase sensitivity.

Response: We have performed a new longitudinal analysis where we compared MindGlide, longitudinal SAMSEG (which explicitly uses longitudinal dimension for increased sensitivity) and WMH-Synthseg. Please see our response to Comment 11 by Reviewer #2.

Comment 14: - again, there seems to be a tendency to cherry-pick results. For instance, fig 6 highlights a lack of sensitivity in synthseg in measuring lesion volume in one particular dataset (PPMS), but doesn't repeat the experiment for the twin dataset (SPMS); doesn't analyze or discuss the atrophy measures that are also shown for both methods; and doesn't comment on the threefold discrepancy between FLAIR and T2 for the proposed method's lesion change estimations in the treatment group (synthseg provides much more consistent estimates).

Response: We added to our analysis a longitudinal comparison of MindGlide, longitudinal SAMSEG and WMH-Synthseg. Figure 6 now shows a comparison between the 3 tools separated by lesion volume, cortical grey matter and deep grey matter. This is explained in detail in Comment 11 by Reviewer #2. Ground truth lesion volumes were only available for our PPMS dataset. Therefore, we focused the comparison of the three methods MindGlide, SAMSEG and WMH-Synthseg on this dataset. We have added detailed discussion as below:

Discussion, Page 21, line 510

“When looking at the results of MindGlide and state-of-the-art approaches, MindGlide-derived lesion volumes demonstrated a treatment effect closely aligned with ground truth values, outperforming SAMSEG, which overestimated, and WMH-Synthseg, which underestimated the treatment effect. Additionally, MindGlide revealed larger differences in regional brain volumes between treatment groups compared to the other tools, indicating enhanced sensitivity in detecting subtle changes.”

Comment 15 :Similarly, fig 7 highlights two specific subjects in which synthseg fails but the proposed method succeeds, but without statistics on a larger number of subjects is unclear how representative this figure is -- presumably some subjects could also be found where synthseg succeeds and the proposed method fails?

Response: We agree and have now performed visual inspection to provide statistics on a larger number of subjects. We manually quality-checked all 433 baseline contrasts from 161 subjects in our routine clinical dataset. Among these, 65 contrasts (15%) were identified as failures using WMH-Synthseg, while only 6 contrasts (1%) were unsuccessful with MindGlide. Notably, of the 6 scans that MindGlide failed to segment, only one was successfully processed by WMH-Synthseg; the remaining 5 failed to achieve segmentation with both tools. This observation underscores the limitations of WMH-Synthseg in this context and explains our decision not to perform a comparative analysis among the three methods within our routine-care clinical dataset.

Results, Page 12, Line 280:

“WMH-Synthseg performed better than both SAMSEG and longitudinal SAMSEG on in the PPMS dataset. Therefore, we only used WMH-Synthseg as a comparator to MindGlide for our analysis of the routine-care clinical dataset. We visually inspected WMH-Synthseg and MindGlide in our routine-care clinical dataset to assess gross segmentation failures, which included 433 baseline contrasts from 161 subjects. WMH-Synthseg demonstrated a significant failure rate, particularly in scans exceeding a thickness of 5 mm. Out of the 433 contrasts that we visually assessed, WMH-Synthseg failed to segment 65 (15%) of the scans, whereas MindGlide exhibited a markedly lower failure rate of only 6 (1%). Of the 6 instances where MindGlide was unable to successfully segment the scans, only one was successfully processed by WMH-Synthseg; the remaining five failed with both methods. Figure 7a shows one example where both, MindGlide and WMH-Synthseg successfully segmented a clinical trial scan. Figure 7b shows a scan from our routine-care clinical dataset with a slice thickness of 7 mm, where WMH-Synthseg’s segmentation failed.”

Comment 16: D. In its current form, the manuscript is very difficult to follow because it lacks structure and doesn't always provide the information needed to correctly interpret the results:

- the section entitled "External validation using independent (unseen in training) datasets" (pages 10-12) should be carefully restructured. It fails to describe the cross-sectional experiments where ICCs are computed when the MRI contrast is changed; it

fails to mention how exactly cross-sectional experiments are performed on data this is longitudinal;

Response: We have restructured the cross-sectional experiment section in Methods to mention how exactly they are done for longitudinal data:

Methods, Page 17, Line 394

“We used the first available visit in each of the longitudinal datasets for cross sectional analysis.”

Regarding Intraclass Correlation Coefficient on different MRI contrast, we have now added the following to this section:

Methods, Page 17, Line 400

“Regarding comparison across MRI contrasts, we used intra-class correlation analysis. Segmentation Consistency across Contrasts: In the PPMS trial, the only dataset with PD, T2, T1, and FLAIR contrasts, we assessed the agreement of segmentations for the same brain structures across these contrasts. We used a hierarchical intraclass correlation coefficient (ICC) to account for the fact that these measurements were taken from the same individuals, which introduces inherent correlation (ICC 3). Only in the SPMS trial, both 3D-T1 and 2D-T1 imaging data was available, and we used these scans to calculate ICC between different resolutions.”

Comment 17: it describes (twice!) an experiment whose results are only given in the suppl. material; and it describes three times (!) the same experiment of comparing automatic with manual lesion segmentations, each time providing additional information.

Response: We apologise for repeating how we used manual labelling of lesions to validate our results. We have merged similar sections and revised the Methods as below:

Merged Methods, Page 19, Line 445

“Manual lesion segmentation by expert neuroradiologists is considered the gold standard in MS. We used two open-source lesion segmentation datasets (called MS-30 and ISBI) and assessed cross-sectional performance against manual lesion segmentations (consensus in MS-30, expert rater in ISBI) using lesion volume and voxel-wise spatial metrics (e.g., Dice score, a standard metric for image segmentation overlap). Cross-sectionally, we assessed lesion load and voxel-wise spatial metrics on 50 FLAIR images from 35 patients, and longitudinally, we calculated the intraclass coefficient (ICC) between raters and MindGlide on the ISBI dataset. For more details, please refer to the Supplemental Material.”

Comment 18: - relevant information regarding the datasets that are used is incomplete and often provided piecemeal in different parts of the manuscript. For instance, patient and imaging characteristics are only provided in the results section, and even then important information (such as the image resolution of the various MRI contrasts in the routine-care dataset, or the number of follow-up scans in all three datasets) is missing.

Response: In our revised manuscript, we added detailed information regarding what data was used for each figure (see Comment 7 and 8 from Reviewer #2) and added information about image resolution and number of follow-up MRIs to our Table 1 to provide additional information for each dataset.

Revised Table 1:

Training and model development set			
	RRMS	SPMS	PPMS
Number (percentage of the sample)	1,082 (38%)	1,453 (51%)	336 (12%)
Age (± standard deviation)	37.1 ± 9.8	47.8 ± 8.2	49.8 ± 8.2
Female (%)	70.6%	60.6%	45.8%
Disease duration (IQR)	1.2 (0.3 - 4.5)	11.7 (5.5 - 16.5)	5.0 (2.5 - 10.8)
EDSS (IQR)	2.0 (1.5 - 3.5)	6.0 (5.0 - 6.5)	4.5 (4.0 - 6.0)
External validation set			
SPMS (n=141)			
Age (± standard deviation)	–	51.3 ± 6.3	–

Female (%)	–	69.3%	–
Disease duration (IQR)	–	20.6 (15.3 - 27.5)	–
EDSS (IQR)	–	6.0 (5.5 - 6.5)	–
Follow up in months (SD)	–	26 (7)	–
# of follow up MRIs	–	2	–
MRI resolution	–	2D-T1: 1x1x3mm 2D-T2: 1x1x3mm 3D-T1: 1x1x1mm	–
PPMS (n=699)			
Age (± standard deviation)	–	–	44.6 ± 8.0
Female (%)	–	–	49.8%
Disease duration (IQR)	–	–	1.5 (0.5 - 4.0)
EDSS (IQR)	–	–	4.5 (3.5 - 6.0)
Follow up in months (SD)	–	–	28 (8)
# of follow up MRIs	–	3	–
MRI resolution	–	T1: 1x1x3mm	–

		T2: 1x1x3mm T2-FLAIR: 1x1x3mm PD: 1x1x3mm	
Real world paediatric cohort (n=161)			
Age (± standard deviation)	14.5 ± 0.2	–	–
Female (%)	77.8%	–	–
Disease duration (IQR)	0.2 (0.1 – 1.0)	–	–
EDSS (IQR)	1.0 (1.0 – 1.5)	–	–
Follow up in months (SD)	12 (19)	–	–
Median MRI slice thickness (Min – Max)	3.3mm (0.4 - 8.5) T1: 1.0mm (0.4 – 7.7) T2-FLAIR: 1.0mm (0.4 - 7.2) T2: 4.8mm (1.0 – 8.5)	–	–

Comment 19: - as already mentioned before, it is never clarified which datasets fig 2(c) and fig 8 are based upon. Further sources of unnecessary confusion include: not mentioning that the upper part of fig 3(a-c) is about PPMS and the lower part about SPMS; not mentioning which datasets were used for the PBVC values of fig 3(d) or how how it was computed (the text even mentions SIENA?), and only defining PBVC in a figure caption; and variously changing the direction of the y-axis and the order of results between/across subplots of fig 3 and fig 4.

Response: We added detailed information about what datasets we used to the captions of all figures. The SIENA pipeline was used to compute PBVC values for both of our trial datasets (PPMS and SPMS). We changed the scale of the y-axis between the subplot for lesion volume changes (fig 3a) and the subplots for grey matter volume changes (fig 3b and 3c) because lesion volumes increase over time and grey matter volumes decrease over time in our dataset. So, we used a positive scale for the lesion volumes and a negative scale for the grey matter volumes. In figure 4 we used different scales for the same reason and to display all relevant values.

Methods, Page 18, Line 413

“We calculated the intra-class correlation coefficient (ICC) for percentage brain volume change with MindGlide segmentations and SIENA algorithm³⁴ (PBVC), a key trial outcome measure, across MRI contrasts (FLAIR, T2-weighted, T1-weighted, and PD in the PPMS trial and T1 and T2-weighted MRI in the SPMS trial).”

Caption Figure 2, Page 36, Line 681

*“Caption: (A) Boxplot displaying Lesion Load estimates (mm³) and distributions measured using ground truth manual delineations (grey), MindGlide (blue) and Freesurfer’s SAMSEG (orange). Lesion load estimates between Ground truth and SAMSEG and MindGlide and SAMSEG methods were significantly different (paired t-tests). (B) Boxplot displaying Dice scores, Sensitivity and Precision measurements for both MindGlide (blue) and SAMSEG (orange) delineated lesions. *** $P < 0.001$, ** $P < 0.01$, * $P < 0.05$, $n=50$. In (C) we calculated Spearman’s correlation coefficients for regional brain volumes obtained from MindGlide and Freesurfer’s SAMSEG and WMH-Synthseg against the expanded disability status scale (EDSS). The analysis evaluates correlations of lesion, deep grey matter (DGM), and cortical grey matter (CGM) volumes with EDSS, across FLAIR and T2 MRI contrasts. As a ground truth comparator for the correlation between lesion volume and EDSS we used manually labelled lesions by expert neuroradiologists. For all tested regions and contrasts MindGlide’s output shows on average higher correlations with EDSS scores except for CGM in T2 (although as shown, they are not statistically significantly different across software). Error bars represent 95% CI. For (A) and (B) we used two openly available lesion segmentation datasets as ground truth comparators ($N = 50$, see supplemental methods).^{37,38} For (C) we used the baseline images of our PPMS dataset ($N = 699$). GT = ground truth (manually labelled lesion segmentation by expert neuroradiologists)”*

Caption Figure 3, Page 38, Line 702

“Caption: MindGlide uniquely enables quantifying treatment effects using single MRI contrasts, including those that have never been used for this purpose (e.g., T2-weighted MRI). (A) – (C) shows longitudinal volume changes with results for our PPMS dataset on the top and results for our SPMS dataset on the bottom. (A) illustrates the annual per cent change in lesion volume detected by MindGlide across FLAIR, PD, T1, and T2 contrasts (resolution: 1x1x3mm) for primary progressive MS (PPMS) and secondary progressive MS (SPMS) cohorts, stratified by treatment allocation. Notably, treatment cohorts exhibited a reduction in lesion volume accrual compared to placebo across all contrasts. Boxes display medians and 95% CI. (B) depicts the annualized rate of cortical grey matter (CGM) atrophy. MindGlide successfully differentiated between treatment and placebo groups, demonstrating reduced cortical atrophy across all MRI contrasts in treated patients. This is also the case for atrophy rates in deep grey matter (DGM) as seen in (C). There are no FLAIR and PD contrasts available for the SPMS cohort. (D) shows inter-contrast consistency for percentage brain volume changes (PBVC): High intra-class correlation coefficients (ICC) for percent brain volume change (PBVC) across different MRI contrasts within the PPMS dataset (2D), indicating high inter-contrast consistency. This underscores the segmentation tool's robustness and consistency in detecting neurodegenerative changes across various imaging contrasts. In the SPMS dataset we compared PBVC of 2D-T1 images and 2D-T2 images with an ICC-coefficient of 0.81 [95% CI 0.73 - 0.87]. PPMS: N = 680, SPMS: N= 130.”

Caption Figure 6, Page 42, Line 744:

“Caption: This figure compares derived percentage volume changes per year of MindGlide, longitudinal SAMSEG and WMH-Synthseg for lesion volume, CGM and DGM separated by treatment groups. We used the PPMS clinical trial for this comparison because it was the largest of our datasets and the only one that includes manually segmented lesion volumes by expert neuroradiologists which we used as ground truth. The effect size calculated using MindGlide-derived lesion volume changes is closest to the ground truth. WMH-Synthseg-derived lesion volume change in the placebo group is closest to the ground truth for both FLAIR and T2 images. Ground truth lesion accrual rate was -1.304% per annum in the treatment group and 3.33% per annum in the placebo group. For FLAIR images, MindGlide detected a lesion accrual rate of 0.64% per annum in the treatment group and 5.95% in the placebo group, compared to 1.863% and 12.566% for longitudinal SAMSEG and 0.56% and 3.11% for WMH-Synthseg. With T2 images, MindGlide showed lesion accrual rates of 2.151% and 6.775% for treatment and placebo groups, while longitudinal SAMSEG showed 4.47% and 13.277% and WMH-Synthseg showed 0.359% and 2.813% respectively. The differences between the three tools in measuring CGM and DGM changes are minor compared to lesion volume changes

except for the CGM estimates of longitudinal SAMSEG. Here, especially in T2 images, longitudinal SAMSEG estimates more atrophy in the treatment group (-0.418% p.a.) than in the placebo group (-0.334% p.a.), although these differences are not significant ($p= 0.291$) PPMS dataset. $N = 680$.”

Caption Figure 8, Page 45, Line 781:

“Caption: Consistency of segmented regions or labels across multiple MRI contrasts measured by the intraclass Correlation Coefficients (ICCs). On the left, coloured brain maps depict all 19 brain region labels: CSF (Cerebrospinal Fluid), 3rd and 4th Ventricle, DGM (Deep Grey Matter), Pons, Brainstem, Cerebellar GM (Grey Matter), Temporal Lobe, Lateral Ventricle Frontal Horn, Lateral Ventricle, Ventral DC (Diencephalon), Optic Chiasm, Cerebellar Vermis, Corpus Callosum, Cerebral WM (White Matter), Frontal Lobe GM, Limbic Cortex GM, Parietal Lobe GM, and Occipital Lobe GM, along with MS (Multiple Sclerosis) Lesions. The right side presents ICC values ranging from 0 to 1 for these regions, providing a quantitative measure of the consistency across multiple MRI contrasts. Higher ICC values indicate greater consistency in the measurement of a particular brain region. PPMS dataset, baseline images, $N = 699$.”

Reviewer #3 (Remarks to the Author):

Comment 1: Goebel et al. propose a deep learning model to extract robust MRI biomarkers from single-contrast scans. The model is validated on data from several clinical trials and an observational cohort. The availability of such a large and diverse dataset offers the opportunity to validate the performance of previously published state-of-the-art models which had not been tested on such heterogeneous data, and to rigorously compare them against newly developed tools. Despite the promising setting, I have some concerns (listed below) on the model evaluation methods adopted, and more generally on the actual novelty of the proposed tool against existing state-of-the-art algorithms. The model architecture, augmentation techniques, and training strategy were leveraged from previous literature. Despite using a larger dataset for training, the proposed model seems to only bring a moderate advantage w.r.t. state-of-the-art models that were trained on substantially smaller cohorts of MS patients – except for a few aspects. Dice scores against manual lesion segmentation, for instance, did show a remarkable improvement from state-of-the-art (although on a small validation set), but this information is somewhat submerged by several other less meaningful results which are maybe given too much emphasis.

Response: We thank the reviewer for comments on our manuscript. We have addressed these important points in the comments below. To summarise:

- (1) we have performed a new analysis for a head-to-head comparison of our method with state-of-the-art and a much larger number (680) of ground truth manually corrected lesion segmentations. Our new results (also shown in new Figure 6) show that MindGlide lesion volumes, compared with WHM-SynthSeg and Samseg, were closest to the ground truth. Please refer to Comment 11 by Reviewer #2 for more details on comparison between MindGlide and state-of-the-art methods.
- (2) We have also added relevant information to the Discussion section.

Discussion, Page 23, Line 538:

“While our proposed model demonstrates an advantage over state-of-the-art models trained on smaller cohorts of MS patients, it is important to recognise that these improvements may not be as pronounced when analysing clinical trial scans, which often feature controlled conditions and high-quality imaging. However, the benefits of our model become significantly more evident in the context of routine-care scans, where variability in image acquisition and image resolution can pose substantial challenges. In these cases, our model's enhancements lead to dramatic improvements in lesion segmentation and analysis, thereby offering valuable insights into real-world clinical scenarios.”

Major points:

Comment 2: 1) My main concern is on the claims of the authors when it comes to treatment effect estimation. Estimating a larger effect does not necessarily mean getting closer to the true effect – in fact, the effect may be overestimated. Both MindGlide and state-of-the-art models should be compared against “ground truth”, i.e., the treatment effect as estimated using hand-labelled lesions. Was manual lesion segmentation available for at least a subset of the longitudinal data used in the study? If not, unless a previously estimated “ground truth” treatment effect is available as a target value for validation, I doubt that any meaningful conclusion can be drawn when it comes to the ability of the proposed model to correctly estimate treatment effect. The same concern applies for treatment effects on brain atrophy.

Response: We agree with this comment. In our revised manuscript, we now added a comparison of manually corrected lesions by expert neuroradiologists (performed independently from our research team using previously published results of the PPMS clinical trial) and the two segmentation tools (see revised Figure 6 on the ground truth treatment effect). For a detailed analysis of these results and revisions in the manuscript please refer to Comment 11 by Reviewer #2. To summarise, when analysing treatment effects we saw a similar treatment effect in MindGlide-derived lesion volumes (5.31% difference between treatment groups in FLAIR images, 4.62% in T2 images) compared to the ground truth (4.63% difference between treatment groups). Longitudinal SAMSEG overestimated the treatment effect on lesion volumes (10.70% difference between treatment groups in FLAIR images, 8.81% in T2 images) and WMH-Synthseg underestimated the treatment effect (2.71% difference between treatment groups in FLAIR images, 2.44% in T2 images).

Comment 3: 2) I think the sentence “MindGlide-derived lesion load had a stronger correlation to EDSS than the state-of-the-art” is a bit misleading and should be softened, as the difference between correlation coefficients is non-significant in 3 out of 4 comparisons.

Response: We agree and removed this sentence from the manuscript. In addition to the correlation between software-derived lesion volume and EDSS, we used a ground truth based on expert manual lesion segmentations. The correlation coefficient between this ground truth and EDSS was 0.131 (95% CI: 0.057 - 0.203; $P < 0.001$). In pairwise comparisons between ground truth and the three tested methods only SAMSEG-derived coefficients using T2 images show a statistically significant difference to the ground-truth-derived coefficient ($P = 0.022$). We refer to Comment 7 by Reviewer #2 for further details and for the changes made in our manuscript.

Comment 4: On a related note: it is reasonable to assume that a better lesion segmentation leads to stronger correlation between lesion load and EDSS. If this is correct, the correlation with EDSS should be even stronger for ground truth hand-

labelled lesions – which are taken as the gold standard for lesion segmentation. Can this be checked from the available data? The “ground truth” correlation would also provide a target value for the correlation with EDSS, to be compared against the one obtained with MindGlide and other software.

Response: We agree with this excellent comment and have performed a new analysis, as shown in our revised Figure 2. Again, we refer to a similar comment (Comment 7) by Reviewer #2.

Comment 5: 3) In the proposed mixed-effects models for estimating treatment effect, the region (or lesion) volume is modelled as a linear function of time, with group-specific slopes. In this context, it does not seem correct to me to include ICV as an additive covariate with a global coefficient which is the same across all subjects. Was there a specific hypothesis motivating this choice? I think it would be more correct to directly use the ratio volume/ICV as a dependent variable, so that the group-specific time slope captures the variation in normalised volume, which is comparable across subjects.

Response: Our cohorts included adults and children. Our decision to use ICV as an additive covariate was to account for individual differences at baseline (such as brain reserve in adults and baseline growth in our paediatric cohort) while ensuring the resulting treatment effects were not reflecting changes before inclusion in the study (Sumowski et al., 2014, Neurology). Especially in our paediatric cohort it is difficult to use a volume/ICV ratio due to different growth trajectories of different brain regions (i.e. the median cortical grey matter volume/ICV ratio in a 10-year-old differs from the median cortical grey matter volume/ICV ratio in an 18-year-old) in children (Bethlehem et al., 2022, Nature). Although we expect to get similar results using the reviewer’s suggested volume/ICV ratio, our approach preserves the original volume scale, which is more intuitive and directly relatable to the clinical context. We have added this to the discussion below:

Discussion Page 25, Line 601

“In our mixed-effects models for estimating treatment effects, we included intracranial volume (ICV) as an extra covariate to account for individual differences at baseline and growth trajectories observed in a paediatric cohort.^{48,49} While using the volume-to-ICV ratio as a dependent variable could yield comparable results, our approach maintains the original volume scale, enhancing interpretability and clinical relevance across diverse age groups.”

Minor points:

Comment 6: 1) Computing time is reported for MindGlide but not for existing models.

Response: Thanks to this comment, we have performed a new analysis. SAMSEG took on average 42 minutes to run, while WMH-Synthseg took on average 15 minutes to run. MindGlide only took 37 seconds for a single contrast to run on GPU. However,

MindGlide runs on GPU while we performed our analysis with WMH-Synthseg and SAMSEG on the default setting, which uses CPU. Therefore, these running times are difficult to compare. Additionally, we did not calculate the time for longitudinal SAMSEG because it analyses MRI scans of multiple timepoints of a patient at once which prolongs the computing time.

Model	Runtime (IQR)	GPU / CPU	GPU Memory	Maximum GPU Memory used during Inference
Mindglide	37 seconds (34 - 40)	GPU: NVIDIA Quadro RTX 6000	24 GB	7850 MiB
SAMSEG	42 minutes (28 – 57)	CPU: Intel(R) Xeon(R) Gold 6140 CPU @ 2.30GHz	-	-
WMH-Synthseg	15 minutes (10 – 19)	CPU: Intel(R) Xeon(R) Gold 6140 CPU @ 2.30GHz	-	-

Comment 7: 2) Was inter-contrast consistency evaluated also for state-of-the-art models? This would be another interesting metric for comparing MindGlide against existing models – especially since robust biomarker extraction from any MRI contrast is one of the main motivations reported by the authors for the development of this new model.

Response: We looked at inter-contrast consistency in WMH-Synthseg and SAMSEG as part of the correlation analysis between regional volumes and EDSS. Figure 2c illustrates how correlation coefficients across contrasts show a higher consistency in MindGlide than in WMH-Synthseg and SAMSEG. In T1 images all three models show a similar correlation between grey matter volumes and EDSS. In Figure 6, MindGlide shows more consistent treatment effects across image contrasts than longitudinal SAMSEG and WMH-Synthseg (see Comments 7 and 11 by Reviewer #2).

Comment 8: 3) The WMH-Synthseg algorithm is trained using only synthetic images generated from segmentation labels. In the Synthseg paper, a supervised version of the model, trained using real scans, had been tested. While performing better than

Synthseg when cross-validated in the same dataset, it proved unable to generalise the segmentation when tested on an external dataset. MindGlide is trained on a dataset including both real scans, and synthetic scans generated with SynthSeg. Have you checked whether the inclusion of the real scans actually leads to any advantage in model performance as compared to training only on synthetic scans (as is the case for SynthSeg)?

Response: While various strategies for training the models have been used, we chose the only one that would maximise the diversity of the data seen by the model, and so we included real scans with synthetic scans. As the reviewer mentions, the improved performance is likely due to the data's high diversity rather than any architectural change (both SynthSeg and MindGlide are 3D CNNs). We aimed to provide a ready-made model across MRI sequences and resolutions. The quantitative effects of synthetic vs real scans would require detailed analysis and retraining of our model with heavy computational expenses that are infeasible in the current setting and out of the scope of our study. We have discussed this important point in the Discussion below:

Discussion, Page 24, Line 575

“It is important to note that MindGlide, like WMH-Synthseg, is a 3D convolutional neural network with the same core architecture. The varying performances in segmenting lesions and detecting treatment effects on grey matter structures are due to the diversity of data used to train these models. We used a combination of real and synthetic scans, while previous studies used synthetic or real scans (but not both). The effect of data diversity is well-known in the machine learning community³⁹. However, quantifying improvements caused by various training regimes with real and synthetic data needs further work and was out of the scope of our study.”

Comment 9: Reviewer #3 (Remarks on code availability):

I did not run the code, but it looks clean and well commented. The git repo does provide a readme file with detailed instructions on installation and usage.

Response: We thank the reviewer for checking our code.